# Opsin 3 mediates UVA-induced keratinocyte supranuclear melanin cap formation

Yinghua Lan[1,2], Wen Zeng[1,2], Yu Wang [1,2], Xian Dong[1,2], Xiaoping Shen[1], Yangguang Gu[1], Wei Zhang[1] & Hongguang Lu [1✉]

Solar ultraviolet (UV) radiation-induced DNA damage is a major risk factor for skin cancer development. UV-induced redistribution of melanin near keratinocyte nuclei leads to the formation of a supranuclear cap, which acts as a natural sunscreen and protects DNA by absorbing and scattering UV radiation. However, the mechanism underlying the intracellular movement of melanin in nuclear capping is poorly understood. In this study, we found that OPN3 is an important photoreceptor in human epidermal keratinocytes and is critical for UVA-mediated supranuclear cap formation. OPN3 mediates supranuclear cap formation via the calcium-dependent G protein-coupled receptor signaling pathway and ultimately upregulates Dync1i1 and DCTN1 expression in human epidermal keratinocytes via activating calcium/CaMKII, CREB, and Akt signal transduction. Together, these results clarify the role of OPN3 in regulating melanin cap formation in human epidermal keratinocytes, greatly expanding our understanding of the phototransduction mechanisms involved in physiological function in skin keratinocytes.

[1] Department of Dermatology, Affiliated Hospital of Guizhou Medical University, 550001 Guiyang, Guizhou, P.R. China. [2] These authors contributed equally: Yinghua Lan, Wen Zeng, Yu Wang, Xian Dong. ✉email: hongguanglu@hotmail.com

Melanin is the primary determinant of skin, hair, and eye color. Importantly, melanin plays an essential role in defending the body against harmful ultraviolet (UV) radiation and other environmental challenges[1]. It is well known that melanin is synthesized in melanocytes melanosomes and translocated into keratinocytes, forming caps over the keratinocyte nuclei[2]. The supranuclear melanin caps reside on the side of the nucleus closest to the skin surface, serving as the tanning response[3,4]. And the supranuclear caps protect human keratinocytes from UV-induced DNA damage[5,6]. It is very important to explore the mechanism of melanin cap formation.

The accumulation of melanin in the perinuclear region is a complex process. Some studies have suggested that cytoplasmic dynein and dynactin are involved in this process[3,7]. Cytoplasmic dynein intermediate chains (Dync1i1) is the linkage protein that binds the entire motor complex to the p150$^{Glued}$ (DCTN1) subunit of the dynactin complex and associated membrane-bound organelles[3]. A subsequent study confirmed that the knockout of DCTN1 could inhibit the perinuclear aggregation of melanin granules and reduce the average centripetal displacement[7]. This means that Dync1i1 and DCTN1 together control the position of perinuclear melanin[3,7]. The mechanism of how keratinocytes sense UV radiation and how UV induces and regulates the distribution of melanin at the top of the nuclei to achieve nuclear protection is still unclear.

All living systems on earth use photons from the sun for adaptive advantage. For example, photon detection enables animals to identify objects through vision[8]. Plants and animals also anticipate the daily light-dark cycle using nonvisual pathways to entrain circadian clocks[9]. UV radiation consists of photons that can activate G protein-coupled receptors (GPCRs) in the eye to induce cellular responses by activating different G proteins and downstream effectors[10]. In animals, most light response pathways employ a member of the opsin (OPN) family of GPCRs as a light detector[10,11]. So far, more than 1000 OPNs have been identified in the animal kingdom[12]. These OPNs have evolved to satisfy the particular light requirements of the organisms that express them[13]. In humans, OPNs are originally found in the eye and play a role in both visual and nonvisual functions[14]. Recently, researchers found that OPN is widely distributed in extra-ocular tissues, including the brain, testes, liver, and kidneys[15,16], but their functions are unclear. The lack of functional data on OPN may be due to its unique structure and extensive expression from deep brain regions to surrounding tissues. In particular, more and more published reports also show that UV-sensing systems may be present in human skin tissues and their cells[17–22]. One of those opsins, OPN3, is predominantly expressed in the cells of cutaneous tissue[21,23–27]. Although the physiological and pathological function of OPN3 has been performed in many aspects, such as melanocyte apoptosis[24], skin pigmentation regulation[25,26,28], and even joining in the skin tumor progress[29], whether OPN3 can participate in and regulate the formation of melanin caps in keratinocytes under UV irradiation needs further elucidation.

Here, we show that OPN3 is the key sensor in keratinocytes responsible for supranuclear cap formation induced via UVA. The supranuclear cap formation induced through OPN3 is calcium-dependent and further activates CaMKII, followed by CREB, leading to the phosphorylation of Akt and ultimately to the increase of the Dync1i1 and DCTN1 expression levels. In conclusion, our study provides insights into the molecular mechanisms by which human keratinocytes respond to UVA radiation to induce supranuclear cap formation, which suggests that OPN3 plays a physiological role in the skin response to UV radiation.

## Results

### UVA mediates keratinocyte melanin cap formation through Dync1i1 and DCTN1.

In the skin, melanin is transferred from melanocytes to neighboring keratinocytes, and the melanin accumulates around the nuclei in the form of melanin caps to mitigate UV damage to DNA. UV can induce the formation of melanin caps in keratinocytes[30]. Despite this, little is known about the fate of the transfer of melanin from melanocytes to keratinocytes and the formation of a cap-like distribution of melanin. In order to determine whether UVA can induce the formation of melanin cap, we first used skin explants as a model in vitro to study, observed the melanin particles increased and tended to gather around the nuclei of keratinocytes in irradiated skin explant (Fig. 1a and Supplementary Fig. s1a). Simultaneously, the melanin particles on the nuclei of the keratinocytes were also detected by transmission electron microscopy (TEM) (Fig. 1b). This melanin cap-like phenomenon is mainly located in the basal region of epidermal keratinocytes. Furthermore, we use human primary keratinocytes (HEK) co-cultured with melanocytes (MC) to observe the formation of melanin caps in keratinocytes after UVA irradiation (Supplementary Fig. s1b). In addition, melanin produced by melanocytes was fed into HaCaT (without melanin) (Supplementary Fig. s2a), and the presence of feeding melanin in HaCaT was confirmed by MF staining, indicating successful feeding (Supplementary Fig. s2b). Next, we irradiated the co-culture model of MC and HaCaT irradiated with UVA to observe the formation of melanin caps in HaCaT (Supplementary Fig. s2c).

Next, we further explored the mechanism of UVA-induced melanin cap formation. UV radiation can induce the expression of dynein in melanocytes and mediate melanin transport[31]. Previous researchers confirmed that Dync1i1 and DCTN1 play an important role in the phagocytosis of melanosomes, the perinuclear aggregation, and the formation of nuclear melanin caps in keratinocytes[3,7]. To determine whether Dync1i1 and DCTN1 are involved in UVA-induced melanin cap formation of keratinocytes, in our experiment, we first irradiated HaCaT with different doses of UVA radiation, and the results showed that 3 J/cm$^2$ UVA significantly upregulated Dync1i1 and DCTN1 proteins (Supplementary Fig. s3a) and mRNA expression levels (Supplementary Fig. s3b, c). Moreover, UVA can upregulate the protein and mRNA expression levels of Dync1i1 and DCTN1 in HEK (Supplementary Fig. s3d–f). In skin explants, Dync1i1 was expressed in full layers of epidermal keratinocytes, and DCTN1 was expressed mainly in basal keratinocytes. And higher expression levels of Dync1i1 and DCTN1 were detected in the UVA irradiation group than in the control group (Supplementary Fig. s3g, h).

To prove whether DCTN1 is involved in UVA-induced keratinocytes melanin cap formation, we used four groups of DCTN1-targeted siRNAs to downregulate the level of DCTN1 mRNA in HEK. The RNAi-DCTN1#1 significantly inhibited the expression of DCTN1 in HEK (Supplementary Fig. s4a). And DCTN1 protein levels are also reduced in HEK expressing DCTN1 # 1 siRNA (Supplementary Fig. s4b). Then, we used RNAi-DCTN1#1 HEK co-cultured with MC and RNAi-DCTN1# HaCaT co-cultured with MC. Given UVA irradiation, we observed the forming ability of melanin cap reduced in the experiment group compared with that in the RNAi-control group both in HEK and HaCaT (Supplementary Fig. s4c, d). These results indicated that DCTN1 is required for UVA-induced keratinocyte supranuclear cap formation.

### UVA regulates supranuclear cap formation in keratinocytes via opsin 3

*Opsin 3 in human epidermal keratinocytes and HaCaT can respond to UVA radiation.* Recently, some researchers proposed a new mechanism: UV participates in skin cell physiological response

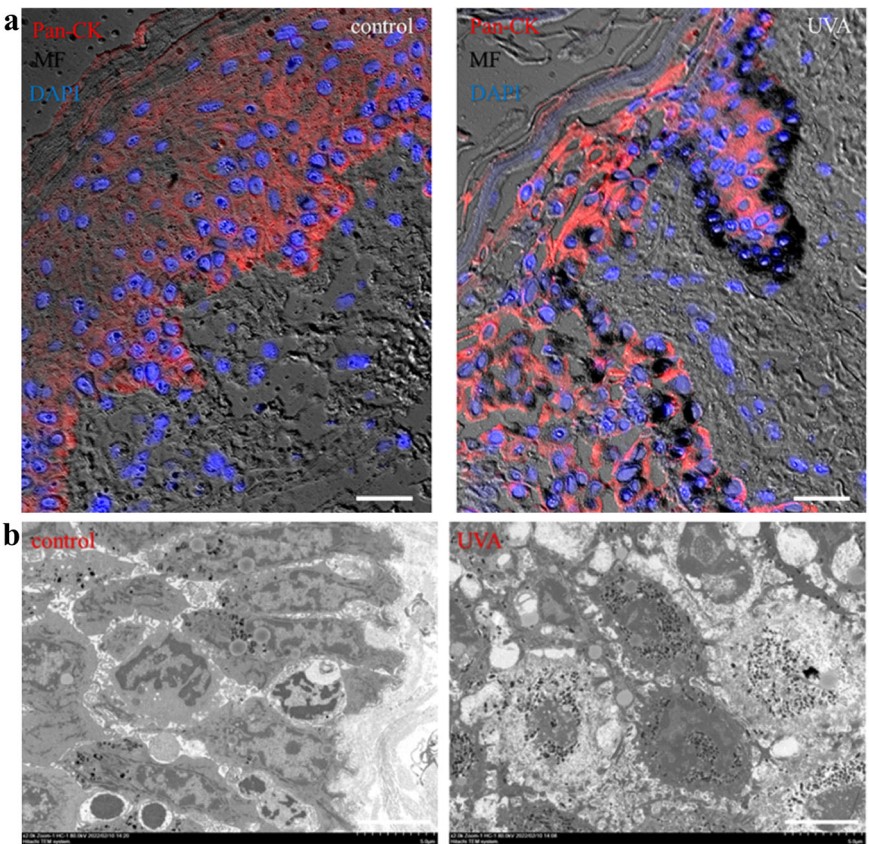

**Fig. 1 UVA induces melanin cap formation in keratinocytes. a** Masson-Fontana (MF) staining showed melanin cap localization in skin explants after 48 h of UVA irradiation (control group (left), UVA-treatment group (right)). Pan-Cytokeratin (Pan-CK) (red) is a marker of keratinocytes. Nuclei were counterstained with 4′,6-diamidino-2-phenylindole (DAPI)(blue). Images were fixed and analyzed by brightfield/fluorescence microscopy. Scale bar = 20 µm. **b** Melanin cap was observed by transmission electron microscopy. Scale bar = 5 µm.

through opsins as a photosensors[19,21,32–34]. Our previous investigation identified the expression of opsins in skin melanocytes[24], nevus cells[25], and fibroblast cells[21]. We further proved opsins mediate appotosis[24], photoaging[21], and melanogenesis[28] in the skin cells. In this study, mRNA expression of OPN (OPN1-sw, OPN2, OPN3, OPN4, and OPN5) was detected in both HEK and HaCaT, and the expression of OPN3 was the highest (Fig. 2a, b). OPN3 protein expression in HEK and HaCaT was also detected (Fig. 2c). Immunofluorescence double staining also appeared positive OPN3 staining on the membrane and cytoplasmic of HaCaT and HEK (Fig. 2d, e). The high expression of OPN3 in the skin keratinocytes suggests that it may function as an epidermal photoreceptor, prompting further studies that might uncover their physiological roles. To determine whether UVA can induce OPN3 expression in HaCaT and HEK, we used different doses of UVA radiation to irradiate HaCaT. The result revealed that 3 J/cm² UVA significantly upregulated the expression of OPN3 (Fig. 2f), and the mRNA level also significantly increased (Fig. 2g). To confirm further these results, we further use 3 J/cm² UVA irradiation on HEK, the protein and mRNA expression levels of OPN3 also significantly increased (Fig. 2h, i). Previous studies have suggested that oxidative damage caused by UVA irradiation may affect melanin response in skin cells[35]. However, in this experiment, the changes of ROS in HEK irradiated with 3 J/cm² UVA were measured by fluorescence microscope and flow cytometry, and these results showed 3 J/cm² UVA could not produce ROS (Supplementary Fig. s5a, b).

*UVA mediates the expression of Dync1i1 and DCTN1 in human epidermal keratinocytes and HaCaT through Opsin 3.* Moreover,

3 J/cm² UVA induced OPN3 expression and melanin cap formation in the skin explant (Fig. 3a). To confirm whether OPN3 is involved in the formation of melanin caps in keratinocytes as a photoreceptor of UVA, we knocked down OPN3 mRNA levels in HEK using 40 nM and 60 nM RNAi-OPN3. It is shown that 60 nM RNAi-OPN3 significantly inhibited the expression of OPN3 (Supplementary Fig. s6a), and OPN3 protein expression levels were also reduced (Supplementary Fig. s6b). Then we used RNAi-OPN3 HEK cultured with MC treated by UVA (3 J/cm²), leading to the formation of melanin cap reduced in the HEK-MC co-culture model (Supplementary Fig. s6c). In addition, we further used LV-OPN3-RNAi to knock down the OPN3 expression of HaCaT by lentivirus transfection technology (Supplementary Fig. s7a–c). We also observed that the formation of melanin caps was reduced in the HaCaT-MC co-culture model (Supplementary Fig. s7d). Furthermore, we inhibited the expression of OPN3 in HEK by siRNA technology and detected no significant changes in the mRNA and protein expression levels of Dync1i1 and DCTN1. We tried to use UVA-irradiated HEK and found that the mRNA and protein expression of Dync1i1 and DCTN1 increased. But, when OPN3 was silent, Dync1i1 and DCTN1 expression induced by UVA was blocked (Fig. 3b–e). We also detected the change of expression level of Dync1i1 and DCTN1 induced by UVA after the knockdown of OPN3 via lentivirus transfection technology in HaCaT. We found that Dync1i1 and DCTN1 mRNA and protein expression levels did not increase (Fig. 4a–d). In addition, we further upregulated the OPN3 expression of HaCaT by lentivirus transfection technology (Supplementary Fig. s7e, f), and the expression of Dync1i1 and DCTN1 increased when treated with UVA (Fig. 4e–h). These results indicate that OPN3 acts as

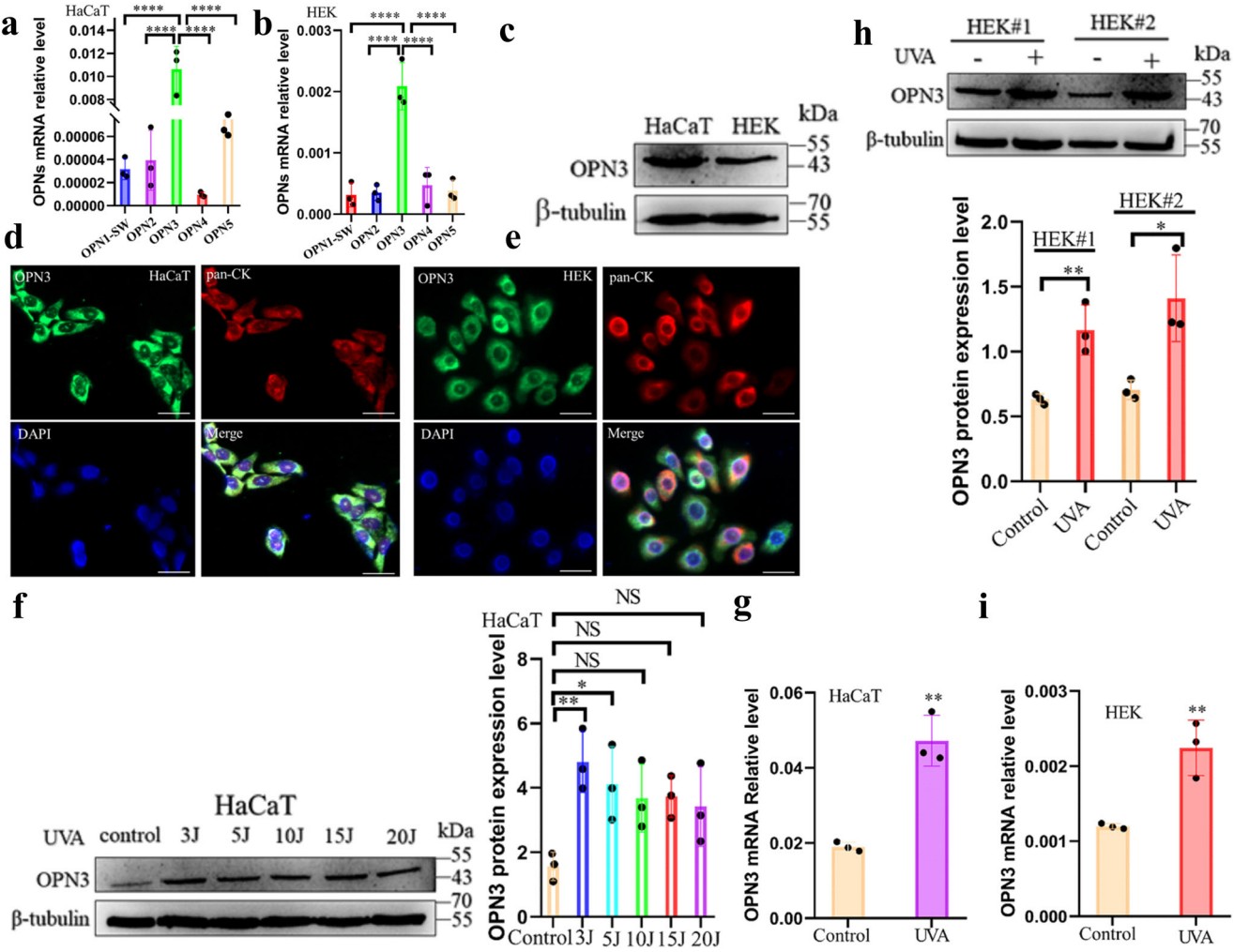

**Fig. 2 UVA increases OPN3 expression in human epidermal keratinocytes. a**, **b** RT-qPCR analysis of OPNs gene expression levels in HaCaT and HEK. OPNs mRNA levels were normalized to GAPDH levels ($n = 3$ independent experiments). Statistical significance was determined by one-ANOVA with post-test. ****$P < 0.0001$. **c** WB analysis of OPN3 protein expression in HaCaT and HEK. **d**, **e** Representative fluorescence confocal images of OPN3 protein expression in HaCaT (left) and HEK (right). OPN3 stained with anti-human OPN3 polyclonal antibody (green), the nucleus stained with DAPI (blue), and the keratinocytes marker Pan-Cytokeratin (Pan-CK) (Cy3, red). Scale bars = 20 μm. Immunofluorescence double staining also appeared positive OPN3 staining on the membrane and cytoplasmic of HaCaT and HEK. **f** HaCaT were irradiated with UVA (3 J cm$^{-2}$, 5 J cm$^{-2}$, 10 J cm$^{-2}$, 15 J cm$^{-2}$, 20 J cm$^{-2}$), and OPN3 protein expression levels were determined by WB analysis. WB analyses were normalized using β-tubulin as a loading control, and the relative protein level was quantified using Quantity One software. $n = 3$ independent experiments. Statistical significance was determined by one-ANOVA with post-test. *$P < 0.05$; **$P < 0.01$. **g** HaCaT were irradiated with 3 J cm$^{-2}$ UVA, and OPN3 mRNA levels were determined by RT-qPCR analysis. OPN3 mRNA levels were normalized to GAPDH levels ($n = 3$ independent experiments). Statistical significance was determined by $t$-test analysis. **$P < 0.01$. **h** HEK were irradiated with 3 J cm$^{-2}$ UVA, and OPN3 protein expression levels were determined by WB analysis. WB analyses were normalized using β-tubulin as a loading control, and the relative protein level was quantified using Quantity One software. $n = 3$ independent experiments. Statistical significance was determined by $t$-test analysis. * $P < 0.05$; **$P < 0.01$. **i** HEK were irradiated with 3 J cm$^{-2}$ UVA, and OPN3 mRNA levels were determined by RT-qPCR analysis. OPN3 mRNA levels were normalized to GAPDH levels. $n = 3$ independent experiments. Statistical significance was determined by $t$-test analysis. **$P < 0.01$.

a key photoreceptor for UVA-induced Dync1i1- and DCTN1-expression-mediated melanin cap formation in keratinocytes.

**UVA mediates Dync1i1 and DCTN1 expression via the OPN3/ calcium /Akt signaling pathway.** Our study and other teams' research showed that ultraviolet or blue light can drive the mobilization of calcium stored in melanocytes and fibroblasts through OPN3[21,26]. Further studies found that UVA led to the intracellular calcium via transient receptor potential channels ankyrin 1 and promoted the phosphorylation of CaMKII in HaCaT[36]. However, whether the calcium signaling pathway is involved in melanin cap formation is unknown. First, we

measured intracellular Ca$^{2+}$ levels in keratinocytes before and after UVA exposure using the fluorometric Ca$^{2+}$ indicator Fluo-3 AM. Our results showed that UVA increased the intracellular Ca$^{2+}$ level (Fig. 5a, b, Supplementary Fig. s8a, b) and upregulated the phosphorylation of CaMKII and CREB (Fig. 5c, d). When OPN3 expression was silent, UVA-induced influx of Ca$^{2+}$ decreased through the immunofluorescence and flow cytometry detecting technology in HEK (Fig. 5e, Supplementary Fig s8c), and we found that after UVA exposure, upregulation expression of the above calcium-related proteins was restrained after decreased expression of OPN3 (Fig. 5f). To test whether Dync1i1 and DCTN1 expression is mediated by calcium signaling via OPN3, we used UVA-exposed HEK treated with PTX, an

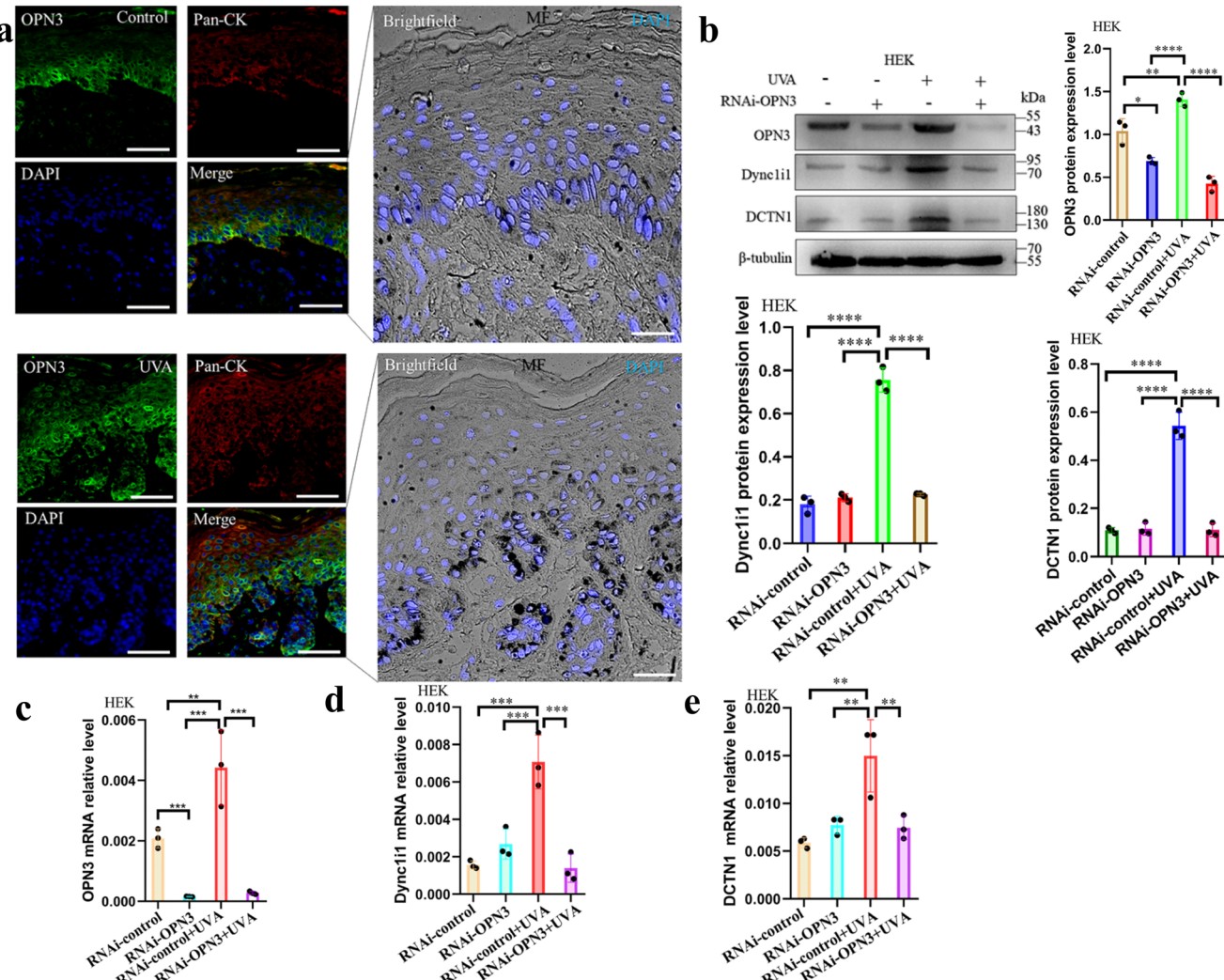

**Fig. 3 UVA induces keratinocyte supranuclear melanin cap via OPN3. a** OPN3 expression (green) colocalized with HEK marker Pan-CK (red) in skin explant with immunofluorescence staining, without UVA (top) or with UVA irradiation (bottom). Nuclei were counterstained with DAPI. Masson-Fontana (MF) staining demonstrated melanin cap formation. Images were analyzed by brightfield/fluorescence microscopy. Scale bar = 20 μm. **b** After siRNA inhibited OPN3 irradiated without or with UVA, WB was used to analyze changes in OPN3, Dync1i1, and DCTN1 expression levels in HEK. WB analyses were normalized using β-tubulin as a loading control, and the relative protein level was quantified using Quantity One software. $n = 3$ independent experiments. Statistical significance was determined by one-ANOVA with post-test. $**P < 0.01$, $****P < 0.0001$. **c** After siRNA inhibited OPN3 irradiated without or with UVA, RT-qPCR was used to analyze changes in OPN3 mRNA levels in HEK ($n = 3$ independent experiments). The data were presented as mean ± SEM. Statistical significance was determined by one-ANOVA with post-test. $**P < 0.01$, $***P < 0.001$. **d** After siRNA inhibited OPN3 irradiated without or with UVA, RT-qPCR was used to analyze changes in DCTN1 mRNA expression levels in HEK ($n = 3$ independent experiments). Statistical significance was determined by one-ANOVA with post-test. $***P < 0.001$. **e** After siRNA inhibited OPN3 irradiated without or with UVA, RT-qPCR was used to analyze changes in Dync1i1 mRNA expression levels in HEK ($n = 3$ independent experiments). Statistical significance was determined by one-ANOVA with post-test. $**P < 0.01$.

inhibitor of Gαi signaling. This UVA-induced intracellular $Ca^{2+}$ reaction was blocked by PTX (Fig. 6a, Supplementary Fig. s8d), and UVA exposure led to increased p-CaMKII and p-CREB levels. This effect was reduced when Gαi was inhibited (Fig. 6b). Since G proteins can cause $Ca^{2+}$ release via phospholipase Cβ PLCβ (PLA) activation[21], we tested whether UVA induced PLC expression. UVA upregulates the expression of PLC (Fig. 6c), and this effect was reduced when Gαi was inhibited (Fig. 6d). In addition, UVA-induced PLC expression was decreased when OPN3 was inhibited (Fig. 6e). These results indicate that UVA regulates the expression of Gαi/ calcium through OPN3. We tested the effect of the PLC antagonist U73122 on UV-induced $Ca^{2+}$ responses. Treatment of HEK with U73122 significantly inhibited UVA-induced $Ca^{2+}$ transients (Fig.6f, Supplementary

Fig. s8e). Simultaneously, although UVA exposure increased p-CAMKII, p-CREB, Dync1i1, and DCTN1 expression levels, this effect was blocked when $Ca^{2+}$ release was inhibited (Fig. 6g). These results indicate that UVA regulates the expression of PLCβ/ calcium through OPN3.

The Akt pathway controlled microtubule formation, likely by regulating accessory protein binding, including that of the dynactin subunit DCTN1[37]. Intracellular calcium mobilization can activate the Akt signaling pathway[38]. Whether Akt plays a role in regulating DCTN1 in human keratinocytes remains unclear. UVA induced Akt phosphorylation (Fig. 7a), and this effect was inhibited when OPN3 expression was silenced (Fig. 7b). Furthermore, UVA-induced Akt phosphorylation was eliminated when $Ca^{2+}$ release was blocked (Fig. 7c). These results suggest that UVA regulates Akt

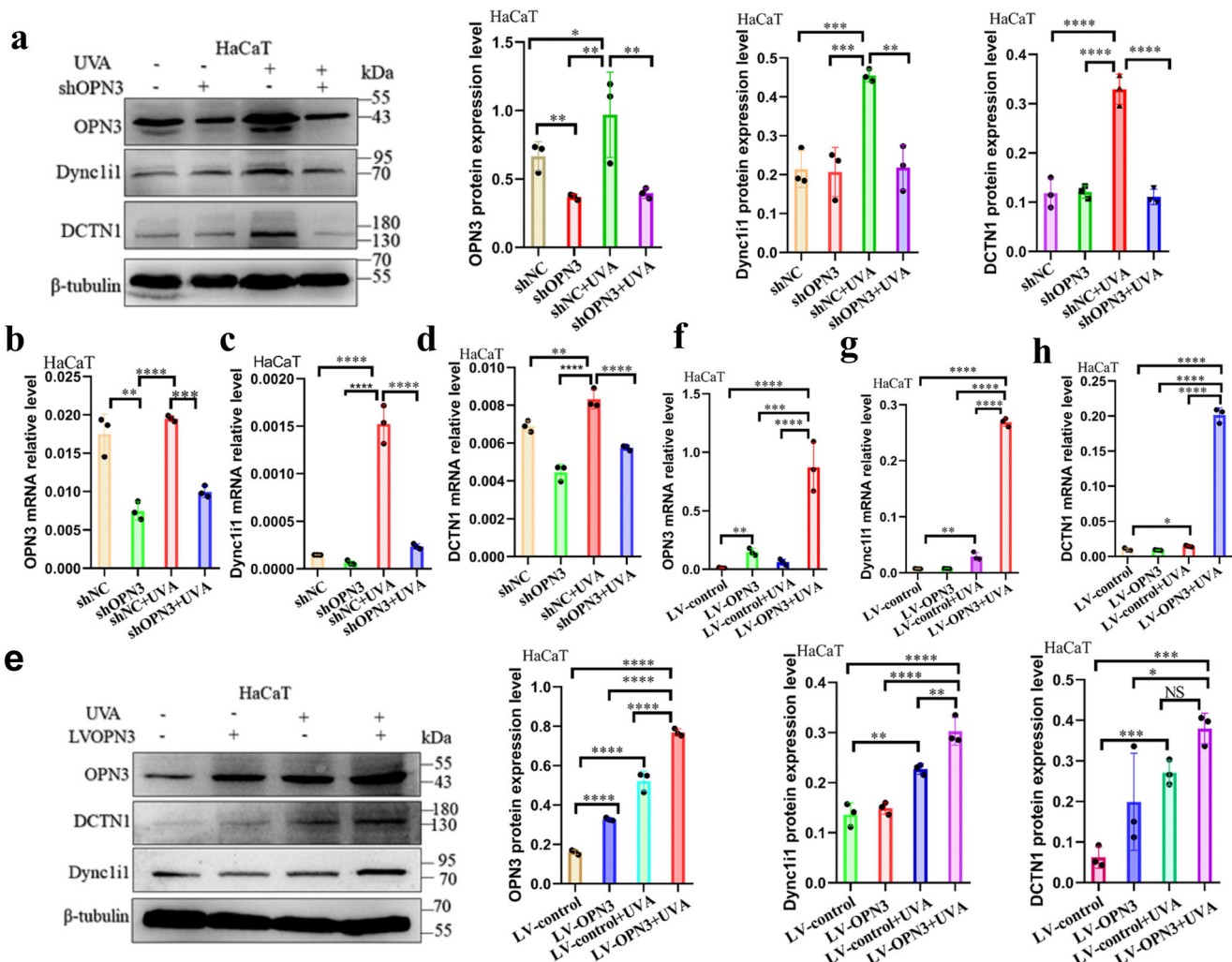

**Fig. 4 UVA upregulates the expression of Dync1i1 and DCTN1 via OPN3. a** HaCaT was transfected with lentivirus shOPN3 (shOPN3#1) and control lentivirus (shNC), and after irradiated without or with UVA, WB was used to analyze changes of OPN3, Dync1i1, and DCTN1 protein expression levels in HaCaT ($n = 3$ independent experiments). WB analyses were normalized using β-tubulin as a loading control, and the relative protein level was quantified using Quantity One software. Statistical significance was determined by one-ANOVA with post-test. *$P < 0.05$, **$P < 0.01$, ***$P < 0.001$, ****$P < 0.0001$. **b–d** After shOPN3 inhibited OPN3 irradiated without or with UVA, RT-qPCR was used to analyze changes of OPN3, Dync1i1, and DCTN1 mRNA expression in HaCaT ($n = 3$ independent experiments). Statistical significance was determined by one-ANOVA with post-test. **$P < 0.01$, ***$P < 0.001$, ****$P < 0.0001$. **e** HaCaT was transfected with lentivirus overexpression OPN3 (LV-OPN3) and control lentivirus (LV-control) and irradiated without or with UVA. Then, WB was used to analyze changes in OPN3, Dync1i1, and DCTN1 expression in HaCaT. β-tubulin was used as a loading control. Statistical significance was determined by one-ANOVA with post-test. *$P < 0.05$, **$P < 0.01$, ***$P < 0.001$, ****$P < 0.0001$. **f–h** After LV-OPN3 overexpression of OPN3 irradiated without or with UVA, RT-qPCR was used to analyze changes in OPN3, Dync1i1, and DCTN1 expression levels in HaCaT. OPN3, Dync1i1, and DCTN1 mRNA levels were normalized to GAPDH levels ($n = 3$ independent experiments). Statistical significance was determined by one-ANOVA with post-test. *$P < 0.05$, **$P < 0.01$, ***$P < 0.001$, ****$P < 0.0001$.

expression through OPN3/ calcium signaling pathway. We further investigated whether Akt is involved in the regulation of DCTN1 expression after UVA irradiation. UVA-regulated Dync1i1 and DCTN1 expression levels were eliminated when Akt phosphorylation was inhibited by MK-2206(2HCl) (Fig. 7d). These results suggest that Akt is involved in the regulation of Dync1i1 and DCTN1 expression.

In summary, our results indicate that UVA induces Dync1i1 and DCTN1 expression to promote melanin cap formation in keratinocytes via OPN3/calcium/Akt signaling pathway (Fig. 8).

## Discussion
In this study, OPN3, as a UVA photoreceptor, plays an important role in the formation of melanin caps in human epidermal keratinocytes. Previous reporters exhibited that the melanin cap deposited in human epidermis keratinocytes acts as a "micro-parasol" to protect the nucleus from UV-induced DNA damage[3,7,39]. The quality and quantity in various racial groups, from Caucasian to black people with skin phototypes, present different melanin content and melanin cap formation in the human epidermis[40,41]. For example, heavily pigmented skin, like black people, harbored more melanin caps than lightly pigmented skin, associated with skin cancer incidence[42,43]. In fact, although melanin cap formation is affected by many factors, it is mainly related to skin color or melanin pigmentation and can be induced by UV irradiation in vivo[30]. However, how UVR or sunlight induces melanin response and melanin particle distribution in human epidermal keratinocytes is not fully understood. Previous studies have used human skin explant models to study the

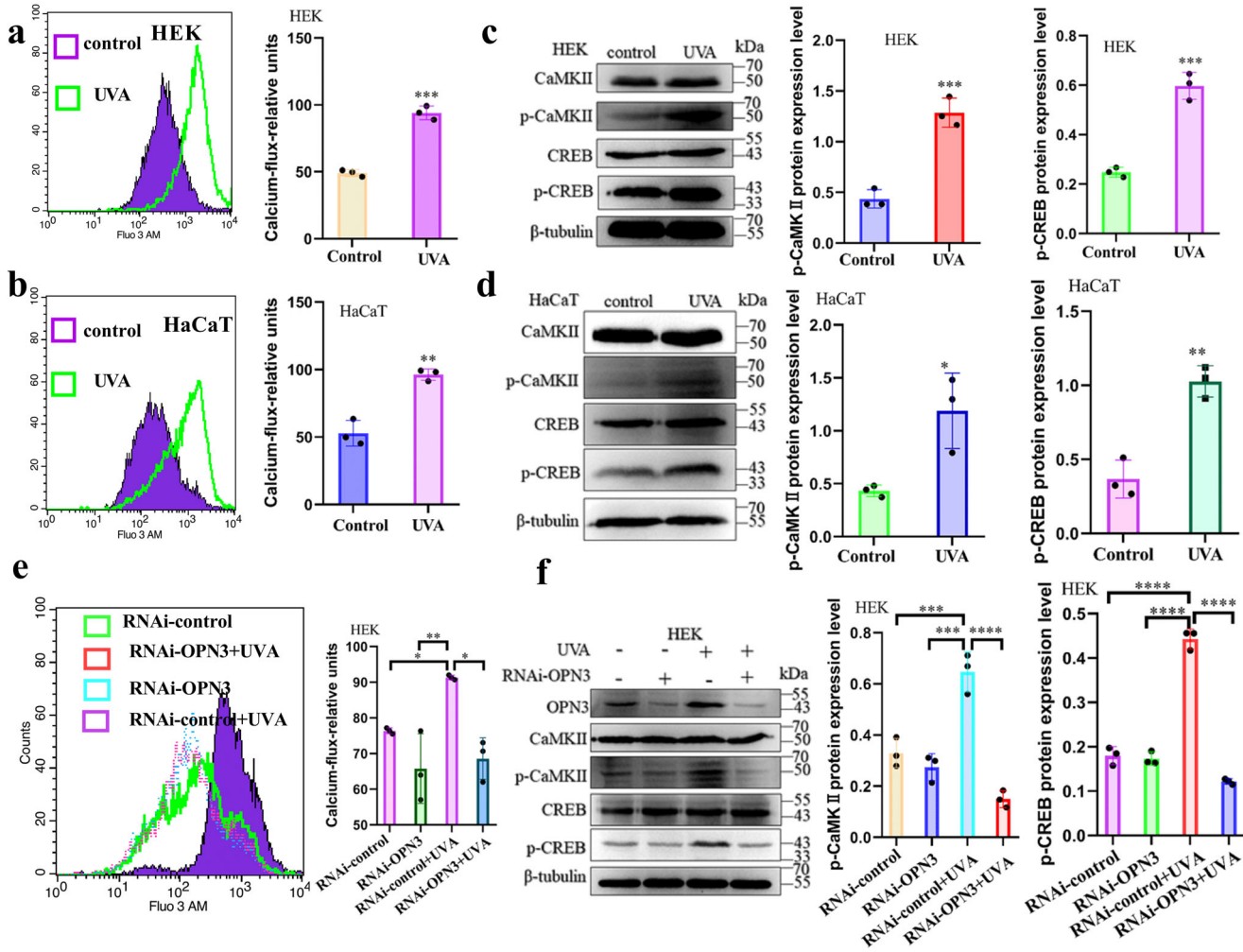

**Fig. 5 OPN3 mediates melanin cap formation via the calcium signaling pathway. a**, **b** HEK and HaCaT calcium fluxes were quantified by flow cytometry. $n = 3$ independent experiments. Statistical significance was determined by $t$-test analysis. **$P < 0.01$, ***$P < 0.001$. **c**, **d** Phosphorylated $Ca^{2+}$/calmodulin-dependent protein kinase (CaMK)II and cyclic adenosine monophosphate response element-binding (CREB) protein were analyzed by WB 1 h after irradiation of HEK and HaCaT with 3 J cm$^{-2}$ UVA. WB analyses were normalized using β-tubulin as a loading control, and the relative protein level was quantified using Quantity One software. Statistical significance was determined by $t$-test analysis. *$P < 0.05$, **$P < 0.01$, ***$P < 0.001$. **e** After siRNA inhibited OPN3 irradiated without or with UVA, calcium fluxes were quantified by flow cytometry. $n = 3$ independent experiments. Statistical significance was determined by one-ANOVA with post-test. *$P < 0.05$, **$P < 0.01$. **f** After siRNA inhibited OPN3 irradiated without or with UVA, WB was used to analyze changes of OPN3, p-CaMKII, and p-CREB protein expression levels in HEK ($n = 3$ independent experiments). WB analyses were normalized using β-tubulin as a loading control, and the relative protein level was quantified using Quantity One software. Statistical significance was determined by one-ANOVA with post-test. ***$P < 0.001$, ****$P < 0.0001$.

formation of melanin caps[3,6,7,30]. We applied this model and confirmed the formation of melanin caps in UVA irradiation. We also tried to use the innovative co-culture model of melanocytes and keratinocytes and further studied the mechanism of melanin cap formation by UVA irradiation. In this study, our results showed that UVA can induce melanin particles to increase and gather around the nuclei of keratinocytes in skin explants and keratinocytes or HaCaT co-cultured with melanocytes.

Byers et al. demonstrated that cytoplasmic dynein participates in melanosome transport in human melanocytes[31]. Cytoplasmic dynein is an ATPase mechanochemical motor that ratchets along microtubules towards the centriolar region[3]. Associated light intermediate cytoplasmic dynein chains and intermediate chains proved to be involved in binding to the dynactin complex that attaches to membrane-bound organelles[44]. Subsequent studies confirmed that Dync1i1 is the motor to partake in the perinuclear-directed aggregation of phagocytosed melanosomes in human keratinocytes[3]. Dynein can be activated when it is

combined with dynamic activating protein. Further studies by Byers et al. confirmed that DCTN1 plays a key role in the formation of UV-induced keratinocytes supranuclear caps[7]. Our data suggest that UVA can provoke the expression of Dync1i1 and DCTN1. Furthermore, when DCTN1 is silenced, UVA-induced melanin cap formation is inhibited. This indicates that DCTN1 plays a significant role in UVA-induced keratinocyte melanin cap formation.

So, how do human keratinocytes detect and respond to UV radiation and then initiate downstream signaling pathways that regulate dynein-mediated melanin cap formation? UV radiation consists of photons. Most light detectors in mammals are OPN, a class of G protein-coupled receptors that convert the energy of a photon into a cellular signaling response[45]. To date, more than a thousand opsins have been identified in animals[13]. Human OPNs can be traditionally classified as visual opsins (OPN1 and OPN2), nonvisual opsins (OPN3, OPN4, and OPN5), and photoisomerases[46,47]. Recently, OPN, as a key photoconductive

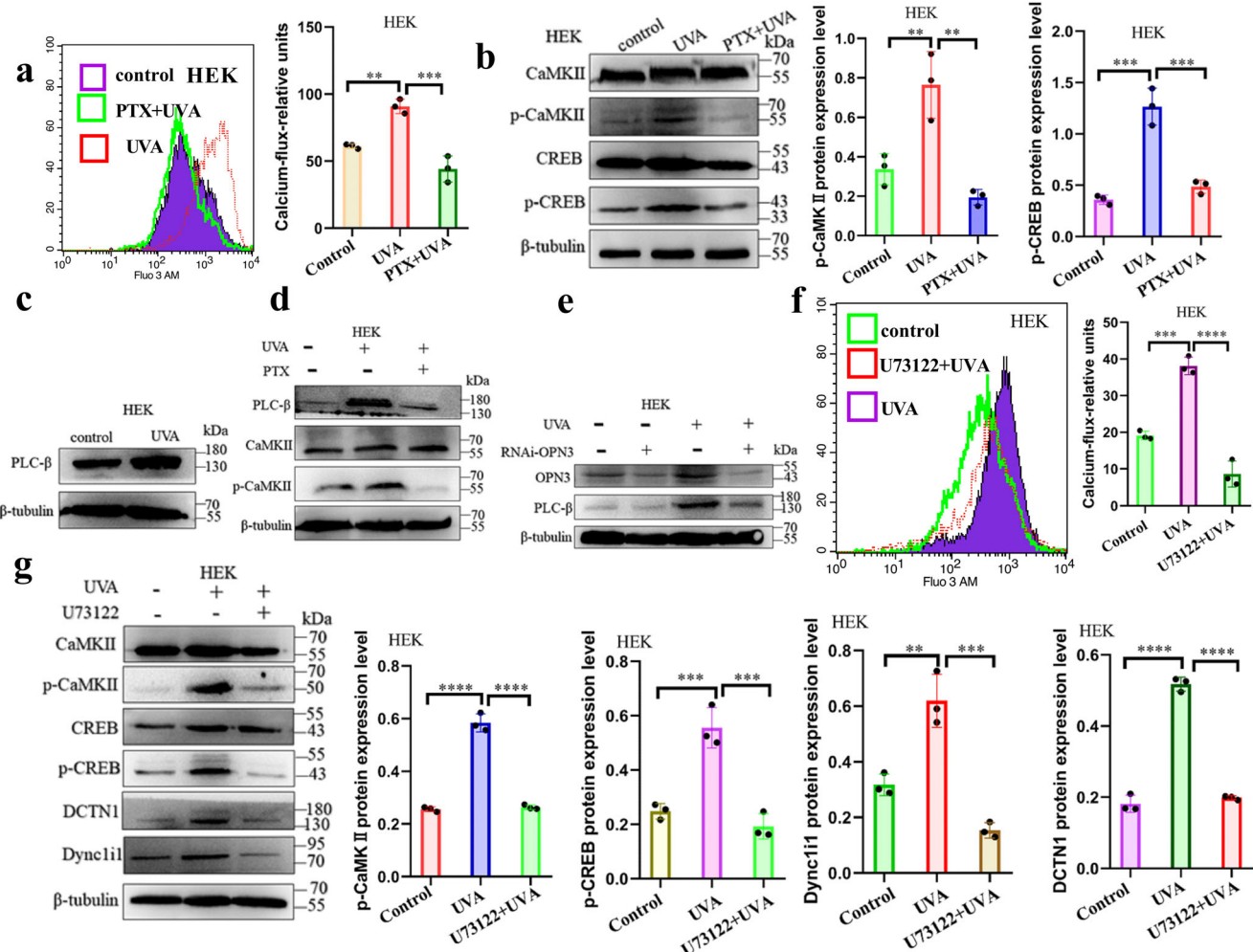

**Fig. 6 OPN3 mediates melanin cap formation via the PLC-β signaling pathway. a** HEK treated with PTX was stimulated with 3 J cm$^{-2}$ UVA, and calcium flux was quantified by flow cytometry. $n = 3$ independent experiments. Statistical significance was determined by one-ANOVA with post-test. **$P < 0.01$, ***$P < 0.001$. **b** HEK treated with PTX were stimulated with 3 J cm$^{-2}$ UVA, and p-CaMKII and p-CREB protein expression level was analyzed by WB. WB analyses were normalized using β-tubulin as a loading control, and the relative protein level was quantified using Quantity One software. Statistical significance was determined by one-ANOVA with post-test. **$P < 0.01$, ***$P < 0.001$. **c** HEK were irradiated without and with UVA, and PLC-β protein levels were determined by WB. **d** HEK treated with PTX were stimulated with 3 J cm$^{-2}$ UVA; protein level was analyzed by WB. **e** HEK were transfected with siRNA against OPN3 irradiated with UVA and lysed after 1 h. Lysates were analyzed by WB using the indicated antibodies (anti-OPN3 and anti-PLC-β). β-tubulin was used as a loading control. **f** HEK treated with U73122 were stimulated with 3 J cm$^{-2}$ UVA; calcium flux was quantified by flow cytometry. $n = 3$ independent experiments. Statistical significance was determined by one-ANOVA with post-test. ***$P < 0.001$, ****$P < 0.0001$. **g** HEK treated with U73122 were stimulated with 3 J cm$^{-2}$ UVA; p-CaMKII, p-CREB, DCTN1, and Dync1i1 protein expression level was analyzed by WB. WB analyses were normalized using β-tubulin as a loading control, and the relative protein level was quantified using Quantity One software. Statistical significance was determined by one-ANOVA with post-test. **$P < 0.01$, ***$P < 0.001$, ****$P < 0.0001$.

molecule found in the retina, has emerged as a previously undiscovered photosensitive element in the skin[23–25]. Other studies and our research work showed that opsins (OPN1-5) are present in human skin[21,23,24]. In this study, OPN3 was highly expressed in human skin keratinocytes and HaCaT. Previous reports showed that OPN3 can mediate melanocytes' melanin synthesis[26,28]. However, whether the distribution of the pigments in keratinocytes is also regulated via OPN3 has not been reported yet. In this experiment, 3 J/cm$^2$ UVA can upregulate the expression of OPN3 in human skin keratinocytes. We further found that melanin granules gathered around or above the nucleus after 3 J/cm$^2$ UVA irradiation. When OPN3 is silenced, the UVA-inducing melanin redistribution effect is inhibited, which indicates that OPN3 is a key photoreceptor for UVA-induced melanin cap formation.

We next sought to determine how OPN3 mediates the melanin cap formation signaling pathway. Previous studies have

shown that photosensitive pigments activate G protein (including Gi-, Gt-, Go-, Gq-, and Gs-)[15]. Mosquito OPN3 activated the Gαi/o subunits of G proteins in a light-dependent manner[15], and Gβγ subunits, which dissociate from Gαi, activated PLC-β to induce Ca$^{2+}$ response[48]. Human OPN3 acts as a photosensitive pigment and activates Gi- and Go-type G proteins in a light-dependent manner, similar to its homologs[21,26]. Based on the above research, we used PTX to prevent Gαi activation and eliminated the effect of OPN3 on Ca$^{2+}$-mediated Dync1i1 and DCTN1 signaling. Further exploration of the Ca$^{2+}$ signaling pathway is important for advancing our understanding of cap structure formation. Since G proteins can cause Ca$^{2+}$ release via PLC-β activation[49,50], we tested the effect of the PLC antagonist U73122 on UVA-induced Ca$^{2+}$ responses and Dync1i1 and DCTN1 expression levels, and the results proved that UVA can induce Ca$^{2+}$ response through OPN3. It also indicates that the

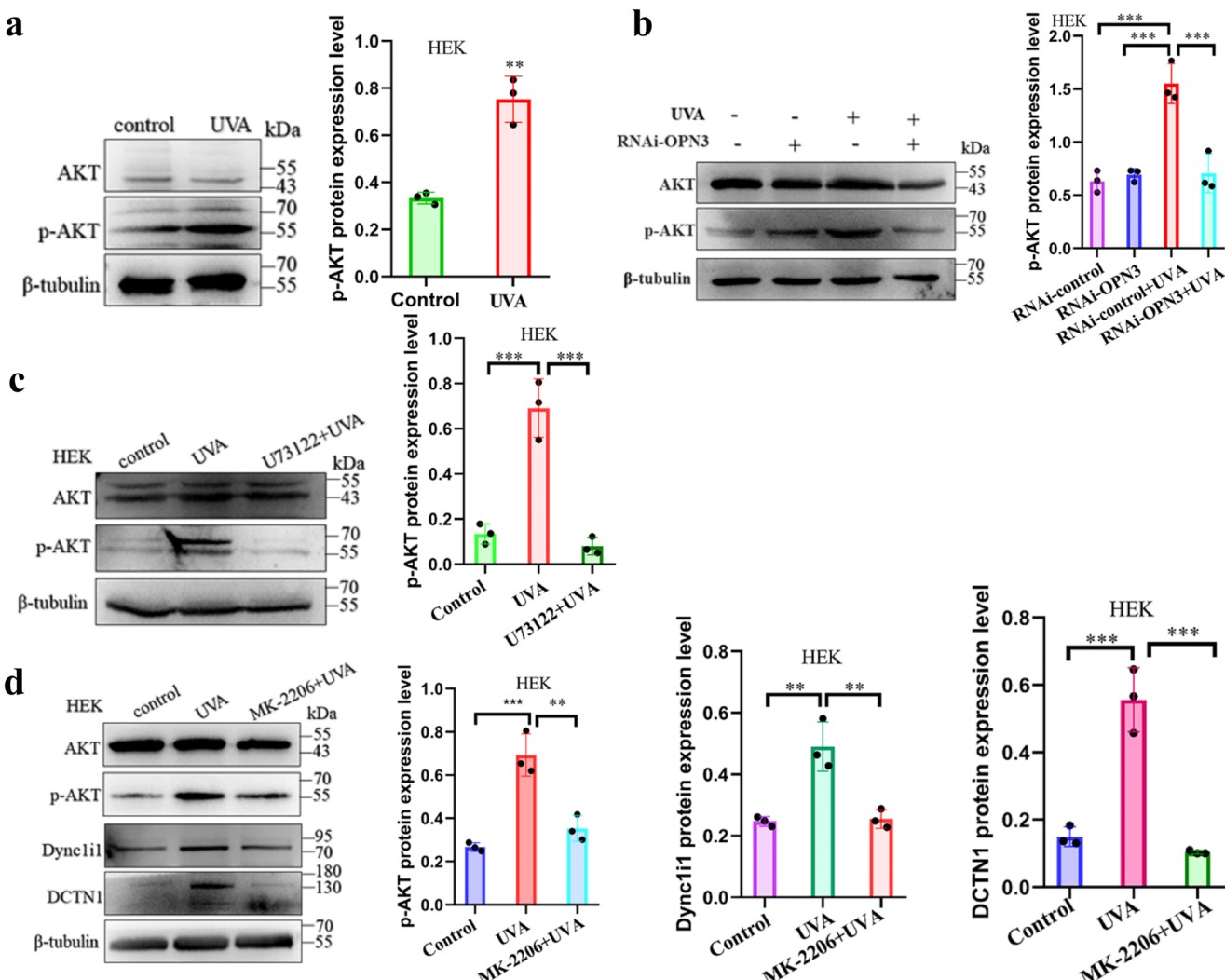

**Fig. 7 OPN3 regulates the expression of Dync1i1 and DCTN1 via the AKT signaling pathway. a** Cells were irradiated without and with UVA, and AKT protein and phosphorylated-AKT (p-AKT) protein levels were determined by WB. β-tubulin was used as a loading control. The relative protein level was quantified using Quantity One software. Statistical significance was determined by t-test analysis. **$P < 0.01$. **b** HEK were transfected with siRNA against OPN3 irradiated with UVA and lysed after 1 h. Lysates were analyzed by WB using the indicated antibodies (anti-OPN3, anti-AKT, anti-p-AKT, anti-Dync1i1, and anti-DCTN1). β-tubulin was used as a loading control. The relative protein level was quantified using Quantity One software. Statistical significance was determined by one-ANOVA with post-test. ***$P < 0.001$. **c** HEK treated with U73122 were stimulated with 3 J/cm² UVA, and protein lysate was analyzed by WB. β-tubulin was used as a loading control. The relative protein level was quantified using Quantity One software. Statistical significance was determined by one-ANOVA with post-test. ***$P < 0.001$. **d** HEK treated with MK-2206 (2HCl) were stimulated with 3 J cm⁻² UVA, and protein lysate was analyzed by WB. β-tubulin was used as a loading control. The relative protein level was quantified using Quantity One software. Statistical significance was determined by one-ANOVA with post-test. **$P < 0.01$, ***$P < 0.001$.

$Ca^{2+}$ signaling pathway plays an important role in melanin cap formation. Some reports have demonstrated that increased intracellular $Ca^{2+}$ activates the Akt signaling pathway[51,52]. Previous research has shown that Akt regulates microtubule formation in fibroblasts[53]. Itoh et al. found Akt signaling pathway contributes to the regulation of DCTN1 in neocortical neurons[33]. In this experiment, we found that calcium mobilization can upregulate the expression of Dync1i1 and DCTN1 by activating the Akt signaling pathway in HEK and HaCaT cells. And up-expression of Dync1i1 and DCTN1 induced by Akt could be inhibited by Akt inhibitor MK-2206.

The findings presented here expand our understanding of OPN3 function and its role as an extra-ocular opsin in human skin. Since OPN3 was shown to be a key sensor for UVA-induced melanin cap formation in human skin keratinocytes, it can be used as a target to protect against DNA damage.

## Methods

**Cell culture.** Human primary keratinocytes (HEK) were derived from prepuce tissue discarded after surgery in children 3–14 years of age. The study was approved by the ethics committee of the Affiliated Hospital of Guizhou Medical University. All subjects' parents or guardians provided written informed consent. Purified HEK was obtained through a two-step enzyme digestion method and was grown in EpiLife® medium (MEPI500CA, Invitrogen Cascade Biologics, USA) with human keratinocyte growth supplement (HKGS, S0015, Invitrogen Cascade Biologics, USA), 2 mM L-glutamine (1051024, Gibco, USA), and a 1% penicillin and streptomycin solution (2114092, Biological Industries, Israel). HEK in the second passage was used in this study. Human immortalized keratinocytes (HaCaT) were purchased from Kunming Cell Bank of Type Culture Collection, Chinese Academy of Science (KCB200442YJ, Kunming, China). HaCaT was cultured in Dulbecco's modified Eagle's medium (DMEM, 0030034DJ, Gibco, USA) containing 10% fetal bovine serum (FBS, FBSSA500-S, AUS GeneX) and a 1% penicillin and strepto-mycin solution and cultured in an incubator with 5% $CO_2$ at 37 °C. HaCaT in passages 10–25 was used in this study.

Human primary melanocytes (MC) were obtained from human children's foreskin with a two-step enzyme digestion method, and MC was immediately

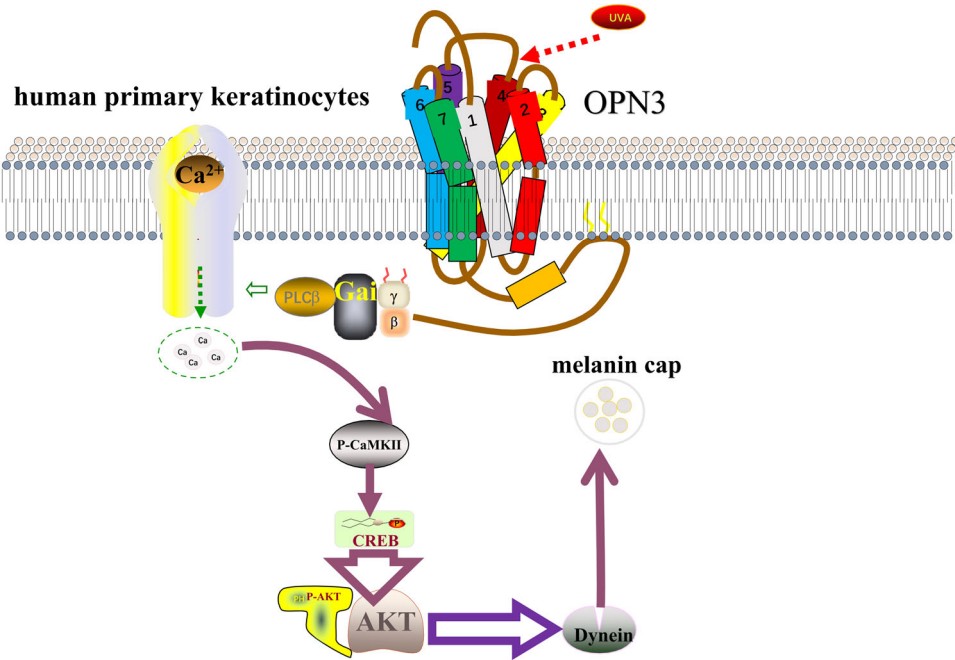

**Fig. 8 Summary model of the key findings in this study.** OPN3 upregulates expression of Dync1i1 and DCTN1 in keratinocytes after UVA radiation exposure via the calcium-dependent G protein-coupled and Akt signaling pathway. (This figure was created based on the elements in the Science Slides software (http://scienceslides.com/).

established in Medium 254(M254500, Gibco, USA) supplemented with a human melanocyte growth supplement (S0165, Gibco, USA), 2 mM L-glutamine, and 1% penicillin and streptomycin solution.

**Skin explant culture.** The removed skin tissues were collected from the Plastic Surgery Department of the Affiliated Hospital of Guizhou Medical University. Skin tissues were transported from surgery to the laboratory in an aqueous 0.9% sodium chloride solution maintained at 4 °C. The skin tissues were cut into pieces of 6 × 7 mm explants within 2 h after surgery. The skin explants were incubated in 12-well plates containing DMEM, 10% FBS, and 1% penicillin and streptomycin solutions for 24 h and then used for subsequent experiments.

**Construction of co-culture models of melanocytes and keratinocytes.** In this study, we established four co-culture cell models: (1) MC and HEK (1:3 ratio) were co-cultured in a mixed medium (EpiLife® medium and Medium 254 at a 1:1 ratio) for 48 h. (2) Knockdown of OPN3 in the HEK using siRNA technology, and then co-culture with MC (1:3 ratio) in a mixed medium (EpiLife® medium and Medium 254 at a 1:1 ratio) for 48 h; knockdown of DCTN1 in the HEK using siRNA technology, and then co-culture with MC (1:3 ratio) in a mixed medium (EpiLife® medium and Medium 254 at a 1:1 ratio) for 48 h. (3) MC and HaCaT (1:3 ratio) were co-cultured in mixed medium (Medium 254 and DMEM at a 1:1 ratio) for 3–5 days before use in subsequent experiments. (4) Silencing of OPN3 in the HaCaT using Lentivirus transfection technology, and then co-culture with MC (1:3 ratio) in a mixed medium (Medium 254 and DMEM at a 1:1 ratio). Knockdown of DCTN1 in the HaCaT using siRNA technology, and then co-culture with MC (1:3 ratio) in a mixed medium (EpiLife® medium and Medium 254 at a 1:1 ratio) for 48 h.

**Chemical preparation and storage.** U73122 (S8011, Selleck Chemicals, Houston, TX, USA) is dissolved in dimethyl sulfoxide (DMSO, D8371, Beijing, China), with the final concentration of 9 mM, and stored at −20 °C. For subsequent experiments, combined with cell viability results (see Supplementary data: Supplementary Fig s9a) and previous research results[49,54], we selected 9 μM U73122 treatment of HEK for 15 min.

MK-2206 (2HCl) (S1078, Houston, TX, USA) is dissolved in dimethyl sulfoxide (DMSO, D8371, Beijing, China), with the final concentration of 12 mM, and stored at −80 °C. For subsequent experiments, combined with cell viability results (see Supplementary data: Supplementary Fig s9a) and previous research results[55], we selected 12 μM MK-2206 treatment of HEK for 24 h.

Pertussis toxin (PTX, P7208, Sigma-Aldrich St., Louis, USA) is dissolved in ddH2O, with the final concentration of 200 μg/ml, and stored at 4 °C. For subsequent experiments, combined with cell viability results (see Supplementary data: Supplementary Fig s9a) and previous research results[48], we selected 200 ng PTX treatment of HEK for 4 h.

Fluo-3 AM (F1241, Invitrogen, Thermo Fisher Scientific, USA) is dissolved in dimethyl sulfoxide (DMSO, D8371, Beijing, China), with the final concentration of 3 mM and stored at the light-free conditions at −20 °C.

**Ultraviolet irradiation (UV).** In this study, an ultraviolet light therapy device (Shanghai Sigma Hi-Tech Co., Ltd., China) was applied. UV dose was measured with an ultraviolet radiometer (Sigma High-Tech Co., Ltd., Shanghai, China). HEK and HaCaT were seeded in 6-well plates at a density of $1.2 \times 10^4$ cells/well. The HEK was irradiated with 3 J cm$^{-2}$ UVA (320–400 nm). The HaCaT was irradiated with different doses of UVA (0 J cm$^{-2}$, 3 J cm$^{-2}$, 5 J cm$^{-2}$, 10 J cm$^{-2}$, 15 J cm$^{-2}$, and 20 J cm$^{-2}$). After 1 h of culture, cells were collected to analyze changes in the protein and mRNA expression levels. Skin explants were cultured in 12-well plates for 24 h and then irradiated with a single dose of 3 J cm$^{-2}$ UVA (320–400 nm). Forty-eight hours later, the explants were collected for analysis (Immuno-fluorescence, Masson-Fontana, and Transmission electron microscopy).

**Supranuclear melanin cap visualization.** The skin explants were biopsied and fixed overnight in neutral-buffered 10% formalin and then embedded in paraffin. They were then applied to Masson-Fontana (MF, G2032, Solarbio, Beijing, China) staining. MF staining was performed according to the manufacturer's instructions. Those cells (co-culture cell models) were then collected and fixed in a 4% paraf-ormaldehyde solution for 15 min and then washed three times in distilled phos-phate buffer solution (PBS) (P1010-2 L, Solarbio, Beijing, China). These samples were incubated in Masson-Fontana melanin staining solution overnight at room temperature (RT). After rinsing three times in distilled water, these samples were incubated in a hypotonic solution for 3 min. Next, the samples were washed in distilled water 5 times. Finally, the samples were visualized under a confocal microscope (Zeiss, Oberkochen, Germany).

**Calcium imaging and calcium flux analysis.** For calcium imaging experiments, cells were collected and incubated with 3 μM Fluo-3 AM for 15 min at 37 °C in the darkroom. Fluorescent images of Fluo-3AM-loaded cells were acquired using Cell Observer-Living Cells (Zeiss, German).

For calcium flux experiments, Fluo-3AM-loaded cells were centrifuged at 1000 rpm for 5 min and once with PBS, and resuspended with 500 μl PBS. The intracellular calcium concentration was detected by flow cytometry (BD Biosciences, San Jose, CA, USA). The excitation source for Fluo-3 AM was a 488-nm air-cooled argon laser, and the emission was measured using a 525-nm band-pass filter.

**Cell immunofluorescence.** HEK and HaCaT were seeded at $1.0 \times 10^4$ cells/well with a cover glass in a 12-well plate and cultured at 37 °C with 5% CO2 for 24 h. The cells were fixed with 95% ethanol at RT for 10 min and then dried at RT. The cells were blocked with 10% FBS for 30 min at 37 °C. After being washed three

times with PBS, the cells were incubated overnight with a rabbit anti-human OPN3 polyclonal antibody (AB_2837240, 1:50, Affinity Biosciences Ltd., Beijing, China) and a mouse anti-human pan-cytokeratin monoclonal antibody (sc-81703, 1:50, Santa Cruz Biotechnology, Inc., USA) at 4 °C. Following washing with PBS three times for 5 min, the cells were incubated for 45 min with Alexa Fluor 488-labeled goat anti-rabbit IgG (A0423; 1:50, Beyotime Biotechnology, Haimen, Jiangsu, China) or Cy3-labeled goat anti-mouse IgG (A0521; 1:50, Beyotime Biotechnology, Haimen, Jiangsu, China) in the dark at 37 °C. Finally, the cell nucleus was stained with DAPI (C0065, Solarbio, Beijing, China) for 10 min at RT. Fluorescent images were collected by fluorescence microscopy (Zeiss, Oberkochen, Germany).

**Skin explants immunofluorescence**. The skin explants were fixed with neutral-buffered 10% formalin overnight, washed with PBS, and then embedded in paraffin. Sections were cut into 3-µm thick slices, deparaffinized in xylene, hydrated in a graded series of ethanol, and subjected to antigen retrieval by the microwave method with EDTA. The samples were blocked with 10% bovine serum albumin and 0.5% Tween-20 in PBS. The slides were incubated overnight with primary antibodies at 4 °C. Incubation with fluorescence-labeled secondary antibodies was performed for 30 min at 37 °C. The following primary and secondary labeled antibodies were used: rabbit polyclonal to OPN3 N-terminal (ab228748; Abcam) conjugated to Alexa Fluor 488-labeled goat anti-rabbit IgG (A0423; Beyotime); mouse anti-human pan-cytokeratin monoclonal antibody (sc-81703; Santa Cruz Biotechnology, Inc.) conjugated to Cy3-labeled goat anti-mouse IgG (A0521; Beyotime); rabbit anti-human Dync1i1 polyclonal antibody (AB_2846296, Affinity Biosciences, Ltd.) conjugated to an Alexa Fluor 488-labeled goat anti-rabbit IgG (A0423; Beyotime); and rabbit anti-human DCTN1 polyclonal antibody (AB_2838577, Affinity Biosciences, Ltd.) conjugated to Alexa Fluor 488-labeled goat anti-rabbit IgG (A0423; Beyotime). Nuclear staining was performed with DAPI. Fluorescent images were collected by fluorescence microscopy (Zeiss, Oberkochen, Germany).

**siRNA transfection**. Generation of OPN3 or DCTN1 knockdown in cells was performed using siRNA technology according to the manufacturer's protocol. The pooled siRNA oligos targeting OPN3, DCTN1, and negative control groups were purchased from TranSheepBio (Shanghai, China). The silencing efficiency of the above different siRNA sequences was analyzed by RT-qPCR compared with negative control groups in 48 h post-transfection. The siRNA sequences with the strongest silencing efficiency for OPN3 or DCTN1 are used for subsequent research. The cells were seeded in 6-well plates at a concentration of $1.0 \times 10^4$ cells/well. When reaching 60% confluence, cells were transfected using Lipofectamine™ 2000 transfection reagent (11668019, Invitrogen, USA) with a final siRNA concentration of 40 nM or 60 nM. After siRNA transfection, cells were cultured for 48 h for further detection. The mRNA and protein silencing levels of OPN3 or DCTN1 were assessed 48 h post-transfection by RT-qPCR and WB. The OPN3 and DCTN1 siRNA sequences were as follows:

OPN3, 5′-GUCACCUUUACCUUCGUGUUTT-3′;
Control siRNA, 5′- UUCUCCGAACGUGUCACGUUTT-3′;
DCTN1-1, 5′-AUUGAUUCAAUGUCUCCAG-3′;
DCTN1-2, 5′-UUGAAGUCAAAGGUCUCUG-3′;
DCTN1-3, 5′- UAGUUCAAACUUCUCCUGG-3′;
DCTN1-4, 5′- AUUCACAUCCUGUAGAUGG-3′;
Control siRNA, 5′-UUCUCCGAACGUGUCACGUTT-3′.

**Lentiviral infection**. According to the standard lentivirus production protocol, lentiviral particles (LV-OPN3-RNAi (shOPN3#1), LV-OPN3-RNAi (shOPN3#2), LV-control-RNAi (RNAi-control)) were obtained through GV493 vectors in the HEK293T cells. The supernatant of the virus was harvested for the following studies.

According to the standard lentivirus production protocol, lentiviral particles (LV-OPN3(45560-1), LV-control (CON238)) were obtained through GV358 vectors in the HEK293T cells. The supernatant of the virus was harvested for the following studies.

HaCaT were cultured on 12-well plates at a density of $3 \times 10^5$ cells/well at 37 °C with 5% CO$_2$ for 24 h. Lentivirus particles with MOI = 10 were added to the medium to infect HaCaT when the cells were grown to 60% confluence in 12-well plates. After 72 h, the medium was replaced with a fresh medium to continue the culture. When 60% of the cells showed green fluorescence, a selection pressure medium including 5ug/ml puromycin was added. Within 3–4 weeks, puromycin-resistant cell colonies were collected to detect the transfection efficiency of lentiviral infection by Western blot and RT-qPCR.

**Real-time quantitative PCR**. Total RNA was isolated from cultured HEK or HaCaT using TRIzol (Invitrogen, Carlsbad, CA) and reverse-transcribed with a Fasting cDNA Dispelling RT SuperMix reverse transcriptase kit (KR118, TIAN-GEN) according to the manufacturer's instructions. Quantitative real-time reverse transcriptase PCR was performed using the Real-time PCR System (Bio-Rad, San Francisco, USA) with SYBR Green PCR Master Mix (Tiangen Biotech, Beijing, China). The relative RNA expression of OPN3, DCTN1, and Dync1i1 was

calculated using the $2^{-\Delta\Delta Ct}$ method, and human GAPDH was used as an internal control. The following human primers were used in this study:

OPN3 forward, 5′ -CAATCCAGTGATTTATGTCTTCATGATCAGAAAG-3′, and
OPN3 reverse, 5′-GCATTTCACTTCCAGCTGCTGGTAGGT-3′;
GAPDH forward, 5′-GACATCCGCAAAGACCTG-3′, and
GAPDH reverse, 5′-GGAAGGTGGCAGCGAG- 3′;
DCTN1 forward, 5′-TCCTGGAGAAGAAGTTGGA-3′, and
DCTN1 reverse, 5′-GATGTCAGCCTGGAGTGC-3′; and
Dync1i1 forward, 5′, -GGTCTGCTTCGCCCGTTTC- 3′, and
Dync1i1 reverse, 5′-CCGCTGCACGGAGTCCTTC-3′. The data were calculated using the $2^{-\Delta\Delta Ct}$ method.

**Western blot analysis**. HEK and HaCaT were lysed with RIPA lysis buffer (R0010, Solarbio, Beijing, China) containing 1% phenylmethylsulfonyl fluoride (PMSF) (P0100, Solarbio, Beijing, China). The extracted total proteins (40 µg) were separated by 10% SDS–PAGE and then transferred to polyvinylidene difluoride (PVDF) (IPVH00010, Merck KGaA, Darmstadt, Germany). The PVDF membranes were blocked with 5% non-fat milk in TBST (T1085, Solarbio, Beijing, China) for 2 h at RT and then incubated with the primary antibodies at 4 °C overnight.

The primary antibodies were used as follows:
Rabbit polyclonal anti-human OPN3 (1:1000, AB_2837240, Affinity Biosciences Ltd., Beijing, China), Rabbit polyclonal anti-human DCTN1 (1:1000, AB_2838577, Affinity Biosciences Ltd., Beijing, China), Rabbit polyclonal anti-human cytoplasmic dynein 1 intermediate chain 1 (Dync1i1, 1:1000, AB_2846296, Affinity Biosciences Ltd., Beijing, China), Rabbit polyclonal anti-human calmodulin-dependent protein kinase II (CaMKII, 1:1000, ab126789, Abcam, Cambridge, UK), Rabbit polyclonal anti-human p-CaMKII (1:1000, ab124880, Cambridge, UK), Rabbit polyclonal anti-human CREB (1:1000, ab32515, Abcam, Cambridge, UK), Rabbit polyclonal anti-human p-CREB (phosphorylated at S133) (1:1000, ab220798, Abcam, Cambridge, UK), Rabbit polyclonal anti-human Akt (1:1000, ABP0059, Abbkine Scientific, Wuhan, China), Rabbit polyclonal anti-human p-Akt (phosphorylated at Ser473) (1:1000, ABP0030, Abbkine Scientific, Wuhan, China). Mouse monoclonal anti-human beta-tubulin (1:1000, AB_2839425, Affinity Biosciences Ltd., Beijing, China) was used as an internal control. After rinsing with TSBT, the membranes were incubated with the corresponding horseradish peroxidase (HRP)-conjugated secondary antibody (goat anti-rabbit IgG H&L, ab97051, or goat anti-mouse IgG H&L, ab6789, Abcam, Cambridge, UK) at RT for 45 min. The protein bands were visualized with an enhanced chemiluminescence (ECL) kit (KF001, Affinity Biosciences Ltd., Beijing, China) with a Bio-Imaging system (Bio-Rad Laboratories, Inc., California, USA).

**Transmission electron microscopy**. At 4 °C, specimens were fixed overnight with 2.5% glutaraldehyde in 0.1 M PBS (pH 7.4). After washing with PBS three times each for 10 min, specimens were post-fixed for 1 h on ice using 1% osmium tetroxide. Specimens were subsequently dehydrated twice with a series of acetone (50%, 70%, 80%, 90%) in proper order for 10 min each time and dehydrated twice with acetone (100%) for 20 min each time at RT and embedding in Epon resin. After polymerizing at 70 °C overnight, resin blocks were sectioned at 65–70 nm using a UC7 ultramicrotome and a diamond knife, and sections were collected on formvar/carbon-coated copper mesh grids. Sections on grids were post-stained with lead citrate uranium acetate. Images were collected using a Hitachi H-7650 transmission electron microscope equipped with an AMT XR41 M digital camera or an FEI Tecnai G2 Spirit BioTWIN transmission electron microscope equipped with an Olympus-SIS Veleta CCD camera.

**Measurement of intracellular ROS levels**. The intracellular ROS levels were measured using a Reactive Oxygen Species Assay Kit (S0033S, Beyotime Biotechnology, Beijing, China) and according to the requirements of the kit. Cells were seeded in 6-well plates for 24 h and then exposed to 3 J/cm$^2$ UVA. One hour after UVA radiation, the cells were incubated with 2′,7′-dichlorofluorescein-diacetate (DCFH-DA) for 20 min at 37 °C and then observed using fluorescence microscopy (Zeiss, Oberkochen, Germany) and flow cytometry (BD Biosciences, San Jose, CA, USA). ROS fluorescence was assessed using the fluorescein isothiocyanate (FITC) channel, and fluorescent images were visualized through confocal laser scanning microscopy. The excitation source for ROS was a 488-nm air-cooled argon laser, and the emission source was a 525-nm band-pass filter.

**Cell viability assay**. CCK8 method was used to analyze the effects of different concentrations of U73122, MK-2206 (2HCl), and PTX on keratinocytes. Briefly, the cells were seeded in 96-well plates at a density of $1.2 \times 10^4$ cells per well. After 24 h incubation, cells were respectively treated with various concentrations of PTX (100 ng/ml, 150 ng/ml, 200 ng/ml, 250 ng/ml, 300 ng/ml, and 350 ng/ml) for 4 h or U73122 (4.5 µM, 9 µM, 13.5 µM, 18 µM, 22.5 µM, and 27 µM) for 30 min or MK-2206 (2HCl) (3 µM,6 µM, 9 µM,12 µM, 15 µM, 18 µM, 21 µM, and 24 µM) for 2 h. Following the treatment, 10 µl CCK8 was added to each well, and 2 h later, absorbance was tested at a wavelength of 405 nm using a fluorescence spectrophotometer (BD Biosciences, San Jose, CA).

**Statistics and reproducibility**. The data were presented as mean ± SEM and were based on at least three separate experiments. The analysis of variance to compare the means of two or more than two groups was performed by $t$-tests or one-way ANOVA with Tukey's post-test analysis of variance. Differences were considered significant when the $P$ value < 0.05. All analyses were carried out with GraphPad Prism Version 8.0 software (GraphPad Software, San Diego, CA, USA).

**Reporting summary**. Further information on research design is available in the Nature Portfolio Reporting Summary linked to this article.

## Data availability

All source data underlying the graphs and charts presented in this study are shown in the Supplementary Data. All original western blot images are shown in the Supplementary Information (Fig. s10). In addition, we have stored all original data in the Dryad system (https://doi.org/10.5061/dryad.wh70rxwrx).

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

## Author contributions

H.L., Y.L., and W.Ze. designed experiments. Y.L., W.Ze., Y.W., and X.D. performed original experiments. X.S. and Y.G. contributed to various experiments. Y.L., W.Ze., Y.W., X.D., W.Zh., and H.L. analyzed and interpreted data. Y.L., X.D., W.Ze., and Y.W. wrote the manuscript. H.L. reviewed the manuscript and supervised the entire study.

## Competing interests

The authors declare no competing interests.
