## [Peer Review File · Communications Biology]

Reviewers' comments:

Reviewer #1 (Remarks to the Author):

In this study, Lan et al. describe a molecular pathway involved in the intracellular movement of melanin in keratinocytes. Using human primary keratinocytes and HaCaT cell lines, the authors argue that OPN3 is a light sensor in human keratinocytes contributing to melanin movement, through activation of Ca⁺/CAMKII and AKT pathway, which ultimately might promote microtubule formation. While the argument is appealing, the main flaw of this manuscript is that the presented data appears weak, with missing replicates. Additional experiments contributing to sustain the findings are very much needed. Also, the methods section needs to be improved with further details on the experimental procedures. Below, I describe my major concerns.

1. Histograms in Fig. S2F lack error bars, suggesting that this procedure was not reproduced. Because reproducibility is a major concern, it should be noted in the figure legends the number of replicates used for the study. The same observation applies for Fig. 3g, 3f, 3h, 4a, 4b, 4c, 4d. Also, in Fig S4L, the authors refer to n=3, but the error bars and statistics on the quantification data are also missing.

2. In figure S4b, it appears that the efficacy of the OPN3 KD using siRNAs is mild, which is also shown in Fig 2B. However, the effect on DCTN1 expression looks stronger than on OPN3, the target of the siRNA itself. Could the authors provide any insight on this apparent discrepancy?

3. Figure legend for S4d panel indicates that HaCat cells were transfected with "full length DNA sequence of human OPN3". How is that strategy making a short hairpin against OPN3? Also, it is indicated that fluorescence microscopy on Fig S4D shows "knocked down OPN3 expression", when in fact it does not. Along the same lines, in figure legend for S4h, again the "full length DNA sequence of human OPN3" appears, yet I suggest revisiting this description, as this correspond to almost 50Kb of DNA, which is too long for lentivirus packaging. Also, the selected transcript variant should be indicated and discussed. Overall, the experimental procedures for KD with sh, and OE are not clear: the sh sequences, how long was the selection with purine toxin, which concentration was used, what passages were used, whether the toxin was present or not etc. should be indicated. Also, figure S4i is referred in the text as "gene level" (line 418), when in fact is transcript. Together, these inaccuracies and omissions make the results section difficult to follow.

4. In the experiments 3a-e, measurements for calcium flux in the cells where OPN3 expression was silenced are encouraged in order to conclude that UVA induces the Ca²⁺ response via OPN3. Additionally, experimental procedures in Fig. 3f-h should be reinforced with additional approaches oriented to link OPN3 action to Gi signaling and PLA activation in keratinocytes.

5. For all experiments using small molecules as activators/ inhibitors, it should be described how was the dose selected, and the length of the treatment. Experiments observing dose-dependent effects are encouraged to support the results and conclusions.

Reviewer #2 (Remarks to the Author):

The manuscript describes investigations regarding the formation of the supranuclear melanin cap in epidermal keratinocytes after receiving the melanin from melanocytes. While the formation of the cap, especially after UV exposure is well known, the molecular mechanism behind this are largely unknown.

The authors hypothesize that OPN 3 is the critical sensor which initiates cap formation after UV exposure. OPN 3 as an UV/Blue light sensor in melanocytes has been described before but its function

in melanin cap formation in keratinocytes is new.

Unfortunately, the paper has some shortcomings and needs a major revision.

The results of the most important experiments are described in Fig. 1 and Fig. 2 because these experiment should provide evidence for the central hypothesis of the paper, involvement of OPSIN 3 in melanin cap formation. However, the Fontana-Masson staining rather shows more melanin after UV (due to oxidative immediate and persistent pigment darkening) than cap formation. The quality of the histology makes it very difficult to accept that there is OPN 3-dependent cap formation after UV. This has to be shown much more convincingly!

The data shown in Fig. 3 and 4 are more convincingly but not relevant at this stage.

Other points:

The authors should strictly separate Results and Discussion. They should not cite other papers in the result section but stick to a description of their own findings and discuss it later in the discussion section.

The manuscript should be shortened. There is no need to go into details about the spectral absorption of OPSIN 3 and other tissues. This is beyond the scope of the paper.

Reviewer #3 (Remarks to the Author):

Lan et al present evidence demonstrating the role of Opsin 3 in mediating UVA induced keratinocyte supranuclear melanin cap formation.

In general this manuscript provides a compelling set of experiments which support their central claims. However the paper could be improved by addressing the following critiques which list enthusiasm.

Fig 1.

panel a magnification is too low and an enlarged inset is needed to fully appreciate capping. this is done later in the paper but should be done here. It appears that in panel d that all melanin synthesis is prevented by siDCTNI. This does not support the central premise and better images should be provided.

Fig 2

It is difficult to appreciate the capping in panels c and e. panels b and d should have quantitation provided. with error bars to show variation across the N

Fig 3

Panels c, d, g, and I should have quantitation. panel g is cut off. panel e requires quantitation.

Fig 4

thank you for providing quantitation however if multiple N there should be error bars to show variation between experiments.

Fig S2

panel f needs quantitation with error bars to show variation.

Fig S3

panel f needs quantitation with error bars to show variation.

Fig S4

panels d and h are too small to interpret. needs larger images of cells and also type is too small in figures. numerous panels missing errors bars for quantitation.

Dear reviewers,

Thank you very much for kind suggestion. We have revised the paper and provided supplementary data according to the suggestion of your as following.

Reviewer #1 (Remarks to the Author):

1. Histograms in Fig. S2F lack error bars, suggesting that this procedure was not reproduced. Because reproducibility is a major concern, it should be noted in the figure legends the number of replicates used for the study. The same observation applies for Fig. 3g, 3l 3f, 3h, 4a, 4b, 4c, 4d. Also, in Fig S4L, the authors refer to n=3, but the error bars and statistics on the quantification data are also missing.

response: Thank you for your suggestion, and we have revised the deficiencies in the manuscript. We have corrected and added the histogram error bars in Fig. S2F, and also added error bars and statistical data in Fig. 3g, 3l 3f, 3h, 4a, 4b, 4c, 4d and Fig. S4L. As the editorial department proposed to make a large number of changes to the manuscript, we have rearranged the pictures to make the manuscript easier to understand.

The original Fig. s2f is changed to Fig. s3a.

The original Fig. 3f is changed to Fig. 5f.

The original Fig. 3h is changed to Fig. 6b.

The original Fig. 3l is changed to Fig. 6e.

The original Fig. 3g is changed to Fig. 6a.

The original Fig. s4L is changed to Fig. 4e.

Fig. s3

Fig. 4

Fig. 5

Fig. 6

2. In figure S4b, it appears that the efficacy of the OPN3 KD using siRNAs is

mild, which is also shown in Fig 2B. However, the effect on DCTN1 expression looks stronger than on OPN3, the target of the siRNA itself. Could the authors provide any insight on this apparent discrepancy?

response: Thank you for your valuable advice, we did not clearly describe the manuscript before, causing misunderstanding. We corrected the mistakes in the manuscript. Moreover, we repeated our experiment many times.

we knocked down OPN3 mRNA levels in HEK using 40nM and 60nM RNAi-OPN3, it is shown that 60nM RNAi-OPN3 significantly inhibited the expression of OPN3(Fig. s6a), and OPN3 protein expression levels were also reduced (Fig. s6b). we detected Dync1i1 and DCTN1 and found no much changes of expression of mRNA and protein levels in both the molecular. And We tried to use UVA irradiate HEK finding the mRNA and protein expression of Dync1i1 and DCTN1 increased. But, when OPN3 was silent, Dync1i1 and DCTN1 expression induced by UVA was blocked (Fig. 3b-e).

Fig. s6

Fig. 3

3. Figure legend for S4d panel indicates that HaCat cells were transfected with “full length DNA sequence of human OPN3”. How is that strategy making a short hairpin against OPN3? Also, it is indicated that fluorescence microscopy on Fig S4D shows “knocked down OPN3 expression”, when in fact it does not. Along the same lines, in figure legend for S4h, again the “full length DNA sequence of human OPN3” appears, yet I suggest revisiting this description, as this correspond to almost 50Kb of DNA, which is too long for lentivirus packaging. Also, the selected transcript variant should be indicated and discussed. Overall, the experimental procedures for KD with sh, and OE are not clear: the sh sequences, how long was the selection with purine toxin, which concentration was used, what passages were used, whether the toxin was present or not etc. should be indicated. Also, figure S4i is referred in the text as “gene level” (line 418), when in fact is transcript.

Together, these inaccuracies and omissions make the results section difficult to follow.

response: Thank you for your valuable advice, we have tried to describe the specific test method again. In addition, we corrected the gene level in Figure S4i to the mRNA level. Please see the revised manuscript.

Easy siRNA design:

NO.	Accession	Target Seq	CDS	GC%
OPN3-RNAi(shOPN3-1)	NM_014322	gtCTTCATGATCAGAAAGTTT	146..1354	31.58%

OPN3-RNAi(shOPN3-2)	NM_014322	gcCTTATATCGTGATCTGCTT	146..1354	36.84%
control sequence(shNC)		TTCTCCGAACGTGTCACGT		
Description	Homo sapiens opsin 3 (OPN3), mRNA.			

Carrier name: GV493

Component sequence: hU6-MCS-CBh-gcGFP-IRES-puromycin

Carrier map:

Overexpression of OPN3 lentivirus design:

Target gene fragment: _

ACCGGTCGCCACCATGTA**CTCGGGGAACCGCAGCGGCGGCCACGGCTACTGGGACGGCGGGCGGGCCGCGGGCGCTGAGGGGCGGGCGCCGCGGGGACACTGAGCCCCGCGCCCTTTCAGCCCCGGCACCTACGAGCGCCTGGCGCTGCTGCTGGGCTCCATTGGGCTGCTGGGCGTCGGCAACAACCTGCTGGTGCTCGTCCTCTACTACAAGTTCCAGCGGCTCCGACTCCCACTCACCTCCTCCTGGTCAACATCAGCCTCAGCGACCTGCTGGTGTCCCTTCGGGGTCACCTTTACCTTCGTGTCCTGCCTGAGGAACGGCTGGGTGTGGGACACCGTGGGCTGCGTGTGGGACGGGTTTAGCGGCAGCCTCTTCGGGATTGTTTCCATTGCCACCCTAACCGTGCTGGCCTATGAACGTTACATTCGCGTGGTCCATGCCAGAGTGATCAATTTTCCTGGGCTGGAGGGCCATTACCTACATCTGGCTCTACTCACTGGCGTGGGCAGGAGCACCTCTCCTGGGATGGAACAGGTACATCCTGGACGTACACGGACTAGGCTGCACGTGGACTGGAATCCAAGGATGCCAACGATTCCCTTTGTGCTTTTCTTATTTCTTGGCTGCCTGGTGGTGCCCTGGGTGTCATAGCCATTGCTATGGCCATATTCTATATTCCATTCGAATGCTTCGTTGTGTGGAAGATCTTCAGACAATTCAAGTGATCAAGATTTTAAATA**

TGAAAAGAACTGGCCAAAATGTGCTTTTTAATGATATTCACCTTCCTGGTCTGTTGGA
 TGCCTTATATCGTGATCTGCTTCTTGGTGGTTAATGGTCATGGTCACCTGGTCACTCCAA
 CAATATCTATTGTTTCGTACCTCTTTGCTAAATCGAACACTGTATAACAATCCAGTGATTTA
 TGTCTTCATGATCAGAAAGTTTCGAAGATCCCTTTTGCAGCTTCTGTGCCTCCGACTGC
 TGAGGTGCCAGAGGCCTGCTAAAGACCTACCAGCAGCTGGAAGTGAAATGCAGATCA
 GACCCATTGTGATGTCACAGAAAGATGGGGACAGGCCAAAGAAAAAAGTGACTTTCA
 ACTCTTCTTCATCATTTTTATCATCACCAGTGATGAATCACTGTCAGTTGACGACAGCG
 ACAAACCAATGGGTCCAAAGTTGATGTAATCCAAGTTCGTCCTTTGACCGGT

Carrier name: GV358

Component sequence: Ubi-MCS-3FLAG-SV40-EGFP-IRES-puromycin

Clone site: AgeI / AgeI

Carrier map:

Figure S4d: Observe the fluorescence of shOPN3-GFP and shNC-GFP under fluorescence microscope.

Lentiviral infection

According to the standard lentivirus production protocol, lentiviral particles (LV-OPN3-RNAi (80513-1), LV-OPN3-RNAi (80514-1), LV-OPN3-RNAi(80515-1), LV-control-RNAi(CON313)) were obtained through GV493 vectors in the HEK293T cells. The supernatant of virus was harvested for the following studies.

According to the standard lentivirus production protocol, lentiviral particles (LV-OPN3(45560-1), LV-control (CON238)) were obtained through GV358 vectors in the HEK293T cells. The supernatant of virus was harvested for the following studies.

HaCaT were cultured on 12-well plates at a density of 3×10^5 cells/well at 37 °C with 5% CO₂ for 24 h. Lentivirus particles with MOI=10 was added to the medium to infect

HaCaT, when the cells were grown to 60 % confluence in 12-well plates. After 72 hours, the medium was replaced with fresh medium to continue the culture. When 60% of the cells showed green fluorescence, selection pressure medium including 5ug/ml puromycin was added. Within 3 – 4 weeks, puromycin -resistant cell colonies were collected to detect the transfection efficiency of lentiviral infection by Western blot and RT-qPCR.

4. In the experiments 3a-e, measurements for calcium flux in the cells where OPN3 expression was silenced are encouraged in order to conclude that UVA induces the Ca²⁺ response via OPN3. Additionally, experimental procedures in Fig. 3f-h should be reinforced with additional approaches oriented to link OPN3 action to Gi signaling and PLA activation in keratinocytes.

response: Thank you for your valuable advice, we have measured calcium flow in cells with OPN3 silencing. In addition, WB was applied to verify the correlation between OPN3 and Gi signal and PLA signal pathways in keratinocytes. First, UVA can induce the expression of PLC(Fig.6c), but this effect was eliminated when U73122 treatment (Fig.6d). in addition, we used UVA-exposed HEK treated with PTX, an inhibitor of G_{αi} signaling. Although UVA exposure led to increased PLC levels, this effect was weakened when G_{αi} was inhibited (Fig.6g). Then, we measured the UVA-induced response to RNAi-OPN3 in HEK and found that although UVA increased PLC levels, this effect was weakened when OPN3 expression was silenced (Fig.6h).

We also rearranged the pictures to make the manuscript easier to understand. The original Fig. 3a-e is changed to Fig. 5 and Fig. 6. This result shows:

Fig. 5

Fig. 6

5. For all experiments using small molecules as activators/ inhibitors, it should be described how was the dose selected, and the length of the treatment. Experiments observing dose-dependent effects are encouraged to support the results and conclusions.

response: Thank you for your valuable advice. In this study, firstly, we analyzed the effects of different concentrations of PTX on cell viability by CCK8 method. In addition, our review of previous literature shows that 200ng PTX treatment of human pigmented melanoma cells for 4 h can significantly inhibit Gi protein (Ozdeslik RN, Olinski LE, Trieu MM, Oprian DD, Oancea E. Human nonvisual opsin 3 regulates pigmentation of epidermal melanocytes through functional interaction with melanocortin 1 receptor. Proc Natl Acad Sci U S A. 2019;116(23):11508-11517.). Combined with cell viability results and literature reports, we selected 200ng PTX treatment of HEK for 4 h.

In this study, we analyzed the effects of different concentrations of U73122 on cell viability by CCK8 method. In addition, our review of previous literature shows that 9 μM U73122 treatment of cells for 15min can significantly inhibit Ca^{2+} transient (Hou C, Kirchner T, Singer M, Matheis M, Argentieri D, Cavender D. In vivo activity of a phospholipase C inhibitor, 1-(6-((17 β -3-methoxyestra-1,3,5(10)-trien-17-yl) amino) hexyl)-1H-pyrrole-2,5-dione (U73122), in acute and chronic inflammatory reactions. *J Pharmacol Exp Ther.* 2004;309(2):697-704. Wicks NL, Chan JW, Najera JA, Ciriello JM, Oancea E. UVA phototransduction drives early melanin synthesis in human melanocytes. *Curr Biol.* 2011;21(22):1906-11.). Combined with cell viability results and literature reports, we selected 9 μM U73122 treatment of HEK for 15min.

MK-2206, an Akt inhibitor. firstly, we analyzed the effects of different concentrations of MK-2206 on cell viability by CCK8 method. Combined with cell viability results and literature reports(Hirai H, Sootome H, Nakatsuru Y, Miyama K, Taguchi S, Tsujioka K, Ueno Y, Hatch H, Majumder PK, Pan BS, Kotani H. MK-2206, an allosteric Akt inhibitor, enhances antitumor efficacy by standard chemotherapeutic agents or molecular targeted drugs in vitro and in vivo. *Mol Cancer Ther.* 2010, 9(7):1956-67.), we selected 12 μM MK-2206 treatment of HEK for 24 h.

Fig. s9

Reviewer #2 (Remarks to the Author):

The results of the most important experiments are described in Fig. 1 and Fig. 2 because these experiment should provide evidence for the central hypothesis of the paper, involvement of OPSIN 3 in melanin cap formation. However, the Fontana-Masson staining rather shows more melanin after UV (due to oxidative immediate and persistant pigment darkening) than cap formation. The quality of the histology makes it very difficult to accept that there is OPN 3-dependent cap formation after UV. This has to be shown much more convincingly! The data shown in Fig. 3 and 4 are more convincingly but not relevant at this stage.

response: Thank you for your valuable advice. In our experiment, we noticed the melanin particles increased and tend to gather round the nuclear of Keratinocytes in

skin explant tissues after UVA irradiation under light microscope (Fig.1a), further under the transmission electron microscope, we observed the melanin particles on the nuclear of the Keratinocytes in irradiated skin explant tissues (Fig.1b). According to your suggestion, we analyzed the changes of ROS in cells irradiated with $3\text{J}/\text{cm}^2$ UVA using cell flow cytometry. Our results showed that $3\text{J}/\text{cm}^2$ UVA could not produce ROS, and the cap formation was not caused by oxidative stress (Fig.s5).

Fig. 1

Fig. s5

Measurement of intracellular ROS levels

The intracellular ROS levels were measured using a Reactive Oxygen Species Assay Kit (S0033S, Beyotime Biotechnology, Beijing, China) and according to the requirements of the kit. Cells were seeded in 6-well plates for 24 h and then exposed to 3J/cm² UVA. One hour after UVA radiation, the cells were incubated with 2',7'-dichlorofluorescein-diacetate (DCFH-DA) for 20min at 37°C and then observed using fluorescence microscopy (Zeiss, Oberkochen, Germany) and flow cytometry (BD Biosciences, San Jose, CA, USA). ROS fluorescence was assessed using the fluorescein isothiocyanate (FITC) channel and fluorescent images are visualized through confocal laser scanning microscopy. The excitation source for ROS was a 488-nm air-cooled

argon laser and the emission source was a 525-nm band-pass filter.

Other points:

The authors should strictly separate Results and Discussion. They should not cite other papers in the result section but stick to a description of their own findings and discuss it later in the discussion section.

response: Thank you for your valuable suggestions. We have revised the manuscript. Please see the results and discussion section of the revised manuscript.

The manuscript should be shortened. There is no need to go into details about the spectral absorption of OPSIN 3 and other tissues. This is beyond the scope of the paper.

response: Thank you for your valuable advice, we have revised the manuscript to reduce its words. please see the revised manuscript.

Reviewer #3 (Remarks to the Author):

Fig 1.

panel a magnification is too low and an enlarged inset is needed to fully appreciate capping. this is done later in the paper but should be done here. It appears that in panel d that all melanin synthesis is prevented by siDCTN1. This does not support the central premise and better images should be provided.

response: Thank you for your valuable advice. According to your kind suggestion, we have adjusted the image to make the image more clearly. And the picture in Fig. 1d was replaced. The original Fig. 1d is changed to Fig. s4d.

Fig. 1

Fig. s4

Fig 2

It is difficult to appreciate the capping in panels c and e. panels b and d should have quantitation provided. with errors bars to show variation across the N

response: Thank you for your valuable advice, due to the typesetting of the manuscript, the picture was compressed, resulting in the distortion of the picture. And panels b and d are quantified.

The original Fig. 2c is changed to Fig. s7a. The original Fig. 2e is changed to Fig. s7b. The original Fig. 2b is changed to Fig. 3b. The original Fig. 2d is changed to Fig. 4a.

Fig.s7

Fig. 3

Fig. 4

Fig 3

Panels c, d, g, and I should have quantitation. panel g is cut off. panel e requires quantitation.

response: Thank you for your valuable advice, we used error bars to quantify Figure 3 in the manuscript. We have corrected and added the histogram error bars and statistical data in Fig. 3c, 3d, 3g, 3i and 3e. As the editorial department proposed to make a large number of changes to the manuscript, we rearranged the pictures to make the manuscript easier to understand.

The original Fig. 3c is changed to Fig. 5c.

The original Fig. 3d is changed to Fig. 5d.

The original Fig. 3l is changed to Fig. 6f.

The original Fig. 3g is changed to Fig. 6b.

Fig. 5

Fig. 6

Fig 4

thank you for providing quantitation however if multiple N there should be error bars to show variation between experiments.

response: Thank you for your valuable advice, we used error bars to quantify Figure 4 in the manuscript. we have also rearranged the pictures to make the manuscript easier to understand. The original Fig. 4 is changed to Fig. 7. Please see the picture below:

Fig. 7

Fig S2

panel f needs quantitation with error bars to show variation.

response: Thank you for your valuable advice, we used error bars to quantify Figure S2f in the manuscript. We have also rearranged the pictures to make the manuscript easier to understand. The original Fig. s2f is changed to Fig. s3a. Please see the picture below:

Fig. s3

Fig S3

panel f needs quantitation with error bars to show variation.

response: Thank you for your valuable advice, we used error bars to quantify Figure s3f in the manuscript. We have also rearranged the pictures to make the manuscript easier to understand. The original Fig. s3f is changed to Fig. 2f. Please see the picture below:

Fig. 2

Fig S4

panels d and h are too small to interpret. needs larger images of cells and also type is too small in figures. numerous panels missing errors bars for quantitation.

response: Thank you for your valuable advice, we rearranged and counted Figure s4 in the manuscript. The original Fig. s4d is changed to Fig. s6c. The original Fig. s4h is changed to Fig. s6f. Please see the picture below:

Fig. s6

REVIEWERS' COMMENTS:

Reviewer #1 (Remarks to the Author):

The authors have very much improved the article after revision, I just have some comments which should be addressed before publication:

- 1) The authors claim that they used paired t-test analysis to determine statistical differences for all experiments. This is not correct as in most of the panels, there are more than two samples to evaluate. Since t-test compares between two samples, for three or more comparisons the ANOVA with post-test or similar should be used. I also suggest using codes to illustrate significance, such as * $P < 0.05$; ** $P < 0.01$, *** $P < 0.0001$, to increase readability. The actual numbers for the test can be included in a supplementary table, for example, if the authors wish so. Of note, paired t-test should be used when each subject or entity is measured twice (i.e, the same individual measured before and after a treatment). Hence, for western blots or qPCR in cell culture this is not possible, because is not the same sample, as once the cells are harvested, they die and cannot be used, treated and measured again. This is the case for panels on Figure S3 and many others, I encourage the authors to please revise the statistical tests, use unmatched ANOVA with posttest (or unpaired t-test for just two comparisons as in Fig S3b and others), or similar adequate statistical analyses for each experiment.
- 2) Line 395: the authors state that OPN3 staining is in the cell membrane, while the images show it all over the cytoplasm. Could the authors comment on this?
- 3) The authors justify in their responses the selection for the concentration and time of exposure to the small molecules used in their studies. I suggest to briefly add these references and justifications to the methods section.
- 4) In general, the manuscript would benefit from carefully revising syntax and spelling. At this regard, these are some lines I detected that need improvement: 45, 54, 56, 67, 68, 72, 78, 80, 346, 351, 365-367, 371, 391-393, 397-398, 404, 411, 422, 458, 486, 490-492, 510,
- 5) Revise figure captions:
 - FigS1a has incorrect indications for the location of control and UVA treated samples' panels.
 - Fig S2a: the MF staining showed THE ABSENCE of melanin in HaCaT, isn't it?
- 6) Revise the order of the figures: for clarity, figures should appear in the order which are mentioned in the text. I suggest reorganizing for example FS7a, locating it after S6b, as in the text, and so on. Provided that these comments can be solved, I would be positive for recommending this research for publication in Communications Biology.

Reviewer #3 (Remarks to the Author):

The authors have addressed my comments and the paper is significantly improved.

Dear reviewer,

Thank you very much for kind suggestion. We have revised the paper according to your suggestions, as shown below. **(Point-by-point response to your comments (the comments by you are in Black, and our responses are in Red))**.

1) The authors claim that they used paired t-test analysis to determine statistical differences for all experiments. This is not correct as in most of the panels, there are more than two samples to evaluate. Since t-test compares between two samples, for three or more comparisons the ANOVA with post-test or similar should be used. I also suggest using codes to illustrate significance, such as * $P < 0.05$; ** $P < 0.01$, *** $P < 0.0001$, to increase readability. The actual numbers for the test can be included in a supplementary table, for example, if the authors wish so. Of note, paired t-test should be used when each subject or entity is measured twice (i.e, the same individual measured before and after a treatment). Hence, for western blots or qPCR in cell culture this is not possible, because is not the same sample, as once the cells are harvested, they die and cannot be used, treated and measured again. This is the case for panels on Figure S3 and many others, I encourage the authors to please revise the statistical tests, use unmatched ANOVA with posttest (or unpaired t-test for just two comparisons as in Fig S3b and others), or similar adequate statistical analyses for each experiment.

Response: Thank you for your suggestion, and we have revised the deficiencies in the manuscript. We have corrected the error description of the two groups of statistical methods. “Statistical significance was determined by paired t-test analysis” correct to Statistical significance was determined by t-test analysis. We recalculated more than three groups of data. The analysis of variance to compare means of two or more than two groups was performed by t tests or one-way ANOVA with Tukey’s post-test analysis of variance. And we use codes to illustrate significance in manuscript, such as * $P < 0.05$; ** $P < 0.01$, *** $P < 0.001$, **** $P < 0.0001$. Please see the corrected manuscript. In addition, according to your suggestion, the actual number of tests in the article has been sorted out and placed in the supplementary table. Please see the supplementary table 1.

2) Line 395: the authors state that OPN3 staining is in the cell membrane, while the images show it all over the cytoplasm. Could the authors comment on this?

Response: Thank you for your valuable suggestions. We reviewed the picture again and corrected the wrong description. “Immunofluorescence double staining also revealed positive OPN3 staining on the HaCaT and HEK membrane (Fig. 2d, e). ” correct to “Immunofluorescence double staining also appeared positive OPN3 staining on the membrane and cytoplasmic of HaCaT and HEK (Fig. 2d, e). ” Please see our revised manuscript.

3) The authors justify in their responses the selection for the concentration and time of exposure to the small molecules used in their studies. I suggest to briefly add these references and justifications to the methods section.

Response: Thank you for your valuable suggestions. We have added the dosages and reasons for selecting inhibitors and literature in the method section.

Chemical preparation and storage

U73122 (S8011, Selleck Chemicals, Houston, TX, USA) is dissolved in dimethyl sulfoxide (DMSO, D8371, Beijing, China), with the final concentration of 9 mM, and stored at -20 °C. For subsequent experiments, combined with cell viability results (see Supplementary data: Fig s9a) and previous research results^{30,31}, we selected 9 μM U73122 treatment of HEK for 15min.

MK-2206 (2HCl) (S1078, Houston, TX, USA) is dissolved in dimethyl sulfoxide (DMSO, D8371, Beijing, China), with the final concentration of 12 mM, and stored at -80 °C. For subsequent experiments, combined with cell viability results (see Supplementary data: Fig s9a) and previous research results³², we selected 12 μM MK-2206 treatment of HEK for 24 h.

Pertussis toxin (PTX, P7208, Sigma–Aldrich St., Louis, USA) is dissolved in ddH₂O, with the final concentration of 200 μg/ml, and stored at 4 °C. For subsequent experiments, combined with cell viability results (see Supplementary data: Fig s9a) and previous research results³³, we selected 200ng PTX treatment of HEK for 4 h.

Fluo-3 AM (F1241, Invitrogen, Thermo Fisher Scientific, USA) is dissolved in dimethyl sulfoxide (DMSO, D8371, Beijing, China), with the final concentration of 3 mM, and stored at the light-free conditions at -20 °C.

4) In general, the manuscript would benefit from carefully revising syntax and spelling. At this regard, these are some lines I detected that need improvement: 45, 54, 56, 67, 68, 72, 78, 80, 346, 351, 365-367, 371, 391-393, 397-398, 404, 411, 422, 458, 486, 490-492, 510,

Response: Thank you for your valuable suggestions. According to your valuable suggestions, we have corrected the manuscript.

For 45:“And the supranuclear cap protect from human keratinocytes from UV induced DNA damage^{5,6}.” correct to “And the supranuclear caps protect human keratinocytes from UV-induced DNA damage^{5,6}. ”

For 54:“That means that Dync1i1and DCTN1 work together control the strategic position of melanin^{3,7}.” correct to “This means that Dync1i1 and DCTN1 together control the position of perinuclear melanin^{3,7}.”

For 56:“The mechanisms how keratinocytes sense UV radiation to induce and regulate the top distribution of melanin on the nuclear to achieve nuclear protective coverage are still unclear.” correct to “The mechanisms of how keratinocytes sense

UV radiation and how UV induces and regulates the distribution of melanin at the top of the nuclear to achieve nuclear protective is still unclear.”

For 67:“In human being, OPN originally found in the eye, which play a role in visual and nonvisual function¹⁴.” correct to “In humans, OPNs are originally found in the eye-and play a role in both visual and nonvisual functions¹⁴.”

For 68:“Recently, researcher have found that OPN is a widely distributed in extraocular tissues, including brain, testes, liver and kidneys^{14,15}, their functions are not completely clear.” correct to “Recently, researcher have found that OPN is a widely distributed in extraocular tissues, including brain, testes, liver and kidneys^{14,15}, their functions are not completely clear.”

For 72:“Particularly, more and more publishing reports have also showed that UV sensing system may exist in human skin tissue and its cells like melanocytes, keratinocytes, fibroblasts and nevus cells¹⁶⁻²².” correct to “In particular, more and more published reports also show that UV-sensing system may present in human skin tissues and their cells, such as melanocytes, keratinocytes, fibroblasts and nevus cells¹⁶⁻²².”

For 78,80:“Whether OPN3 can be able to participated and regulated the melanogenesis process, i.e. melanin cap formation in keratinocytes by UV irradiation needs further elucidated.” correct to “Whether OPN3 can participate in and regulate the formation of melanin caps in keratinocytes under UV irradiation needs further elucidation.”

For 346:“Despite all this, little is known about the fate of melanin transferred from melanocytes to keratinocytes in which forming the melanin cap-like distribution.” correct to “Despite this, little is known about the fate of the transfer of melanin from melanocytes to keratinocytes and the formation of a cap-like distribution of melanin.”

For 351:“Simultaneously, transmission electron microscopy also detected that the melanin particles on the nuclear of the keratinocytes (Fig. 1b).” correct to “Simultaneously, the melanin particles on the nuclear of the keratinocytes were also detected by transmission electron microscopy (TEM) (Fig. 1b).”

For 365-367:“Firstly, we used different doses of UVA radiation to irradiate HaCaT,

showed the 3J/cm² UVA significantly induced the expression of Dync11l and DCTN1(Fig. s3a), and this mRNA expression level also confirms this result (Fig. s3b,c).” correct to “First, we irradiated HaCaT with different doses of UVA radiation, and the results showed that 3J/cm² UVA significantly induced Dync11l and DCTN1 proteins (Figure; s3a) and mRNA expression levels (Figure s3b, c).”

For 371:“In skin explant, the fluorescence intensity of Dync11l expressed in keratinocytes in epidermis and DCTN1 in the basal layer keratinocytes, which the UVA irradiation group was significantly higher than control group (Fig. s3g, h).” correct to “In skin explants, Dync11l was expressed in epidermal keratinocytes and DCTN1 was expressed in basal keratinocytes, and was significantly higher in UVA irradiation group than in control group (Fig. s3g, h).”

For 391-393: “In this study, OPN (OPN1-sw, OPN2, OPN3, OPN4 and OPN5) mRNA is expressed in both HEK and HaCaT (Fig. 2a, b). And OPN3 mRNA are expressed at significantly higher levels in both HEK and HaCaT (Fig. 2a, b). OPN3 protein expression in HEK and HaCaT also detected (Fig. 2c).” correct to “In this study, mRNA expression of OPN (OPN1-sw, OPN2, OPN3, OPN4 and OPN5) was detected in both HEK and HaCaT, and the expression of OPN3 was the highest (Figure 2a, b).”

For 397-398: “Whether UVA can induce the expression of OPN3 in HaCaT and HEK.” correct to “Whether UVA can induce OPN3 expression in HaCaT and HEK.”

For 404: “Previous report show that oxidative damage induced by UVA irradiation may affect cutaneous cells’ melanin pigment reaction³¹.” correct to “Previous studies have suggested that oxidative damage caused by UVA irradiation may affect melanin response in skin cells³¹.”

For 411: “Whether OPN3 as a photosensor of UVA participate in melanin cap formation in keratinocyte,” correct to “Whether OPN3 is involved in the formation of melanin caps in keratinocytes as a photoreceptor of UVA,”

For 422: “Further, after inhibiting the expression of OPN3 in HEK using siRNA technology, we detected Dync11l and DCTN1 and found no much changes of expression of mRNA and protein levels in both the molecular.” correct to

“Furthermore, we inhibited the expression of OPN3 in HEK by siRNA technology, and detected no significant changes in the mRNA and protein expression levels of Dync1i1 and DCTN1,”

For 458: “we tested the effect of the PLC antagonist U73122 on UV-induced Ca²⁺ responses.” correct to “We tested the effect of the PLC-β antagonist U73122 on UV-induced Ca²⁺ responses.”

For 486: “The quality and quantity in various racial groups from Caucasian to black people with skin phototypes present different menain content and melanin cap formation in human epidermis.” correct to “The quality and quantity in various racial groups from Caucasian to black people with skin phototypes present different melanin content and melanin cap formation in human epidermis.”

For 490-492: “however, UVR or sunlight how to induced the melanin reaction and melanin particle distributed in human epidermis keratinocytes have not fully understood.” correct to “However, how UVR or sunlight induces melanin response and melanin particle distribution in human epidermal keratinocytes is not fully understood.”

For 510: “Our data suggest that UVA can be able to provoke Dync1i1 and DCTN1 expression.” correct to “Our data suggest that UVA can provoke the expression of Dync1i1 and DCTN1.”

5) Revise figure captions:

-FigS1a has incorrect indications for the location of control and UVA treated samples' panels.

Response: Thank you for your suggestion, and we have revised the deficiencies in FigS1a.

-Fig S2a: the MF staining showed THE ABSENCE of melanin in HaCaT, isn't it?

Response: yes.

6) Revise the order of the figures: for clarity, figures should appear in the order which are mentioned in the text. I suggest reorganizing for example FS7a, locating it after S6b, as in the text, and so on.

Response: Thank you for your valuable suggestions. According to your valuable suggestions, we have adjusted the order of pictures.

Fig. s6

Fig. s7